

# Probing the exchange of CO₂ and O₂ in the shallow critical zone during weathering of marl and black shale

Tobias Roylands[1], Robert G. Hilton[2], Erin L. McClymont[1], Mark H. Garnett[3], Guillaume Soulet[4], Sébastien Klotz[5], Mathis Degler[6], Felipe Napoleoni[7], Caroline Le Bouteiller[5]

[1]Department of Geography, Durham University, Durham, DH1 3LE, United Kingdom
[2]Department of Earth Sciences, University of Oxford, Oxford, OX1 3AN, United Kingdom
[3]NEIF Radiocarbon Laboratory, SUERC, East Kilbride, G75 0QF, United Kingdom
[4]Ifremer, Geo-Ocean, Brest University, CNRS, Plouzané, 29280, France
[5]INRAE, Univ. Grenoble Alpes, CNRS, Grenoble, 38000, France
[6]Institute of Geosciences, Christian-Albrechts-University, Kiel, 24118, Germany
[7]Centro de Estudios Científicos, Valdivia, 5110466, Chile

*Correspondence to*: Robert G. Hilton (robert.hilton@earth.ox.ac.uk)

**Abstract.** Chemical weathering of sedimentary rocks can release carbon dioxide ($CO_2$) and consume oxygen ($O_2$) via the oxidation of petrogenic organic carbon and sulfide minerals. These pathways govern Earth's surface system and climate over

geological timescales, but the present-day weathering fluxes and their environmental controls are only partly constrained due to a lack of in situ measurements. Here, we investigate the gaseous exchange of $CO_2$ and $O_2$ during the oxidative weathering of black shales and marls exposed in the French southern Alps. On six fieldtrips over one year, we use drilled headspace chambers to measure the $CO_2$ concentrations in the shallow critical zone, and quantify $CO_2$ fluxes in real-time. Importantly, we develop a new approach to estimate the volume of rock that contributes $CO_2$ to a chamber, and assess effective diffusive

gas exchange, by first quantifying the mass of $CO_2$ that is stored in a chamber and connected rock pores. Both rock types are characterized by similar contributing rock volumes and diffusive movement of $CO_2$. However, $CO_2$ emissions differed between the rock types, with yields over rock outcrop surfaces (inferred from the contributing rock volume and the local weathering depths) ranging between 166 tC km$^{-2}$ yr$^{-1}$ and 2,416 tC km$^{-2}$ yr$^{-1}$ for black shales and between 83 tC km$^{-2}$ yr$^{-1}$ and 1,558 tC km$^{-2}$ yr$^{-1}$ for marls over the study period. Having quantified diffusive processes, chamber-based $O_2$ concentration

measurements are used to calculate $O_2$ fluxes. The rate of $O_2$ consumption increased with production of $CO_2$, and with increased temperature, with an average $O_2 : CO_2$ molar ratio of 10 : 1. If $O_2$ consumption occurs by both rock organic carbon oxidation and sulfide oxidation, either an additional $O_2$ sink needs to be identified, or significant export of dissolved inorganic carbon occurs from the weathering zone. Together, our findings refine the tools we have to probe $CO_2$ and $O_2$ exchange in rocks at Earth's surface and shed new light on $CO_2$ and $O_2$ fluxes, their drivers and the fate of rock-derived

carbon.



# 1 Introduction

Sedimentary rocks cover ~ 64 % of the present-day continental surface of Earth (Hartmann & Moosdorf, 2012) and contain vast amounts of carbon in carbonate minerals and organic matter (Petsch, 2014). The chemical breakdown of these rocks can act as a source of carbon dioxide ($CO_2$) to the near-surface reservoirs (hydrosphere-biosphere-pedosphere-atmosphere) and

can be a sink of oxygen ($O_2$), in turn exerting an important control on the evolution of climate and life (Berner & Berner, 2012; Berner, 1999; Sundquist & Visser, 2003). Two reactions are recognized: i) the oxidation of petrogenic organic carbon ($OC_{petro}$) (Petsch, 2014); and ii) the oxidation of sedimentary sulfide minerals that produce sulfuric acid that can, in turn, dissolve carbonate minerals (Calmels et al., 2007; Li et al., 2008; Torres et al., 2014). On a global scale, these chemical weathering pathways together emit roughly as much $CO_2$ to the atmosphere as is removed by the weathering of silicate

minerals with a flux of ~ 90 MtC yr$^{-1}$ – 140 MtC yr$^{-1}$ (Gaillardet et al., 1999; Moon et al., 2014). How these fluxes play out over longer timescales remains difficult to assess (Hilton & West, 2020; Petsch, 2014), however the decline of atmospheric $O_2$ over the last 800,000 years (Stolper et al., 2016) has been tentatively linked to changes in global oxidative weathering fluxes (Yan et al., 2021). To improve the understanding of the changes of Earth's surface conditions over geological timescales, the mechanism and controls on oxidative weathering pathways need to be better constrained (Berner & Berner,

2012; Mills et al., 2021). Theoretical modeling of $OC_{petro}$ oxidation currently relies on input kinetics of the weathering reactions from laboratory experiments (Bao et al., 2017; Bolton et al., 2006; Petsch, 2014). In situ gas exchange between rocks undergoing weathering and the atmosphere can provide much needed insight.

The first field-based fluxes of weathering-derived $CO_2$ were reported by Keller & Bacon (1998) in a glacial till dominated by shales. More recently, Tune et al. (2020) found substantial $CO_2$ release and $O_2$ depletion in bedrock

undergoing weathering below a forested hillslope. There, carbon release is mostly sourced from superficial soils, deep roots, with minor contributions from $OC_{petro}$ oxidation (Tune et al., 2020, 2023). At both sites, the gaseous fluxes were determined on the basis of profiles of the $CO_2$ concentration in air sampled from boreholes extending to depths of ~ 7 m (Keller & Bacon, 1998) and of ~ 16 m (Tune et al., 2020), using Fick's law:

$$\dot{j}_X = -D_X \times \frac{\mathrm{d}c_X}{\mathrm{d}z} , \tag{1}$$

where $j_X$ is the molar flux (mol m$^{-2}$ s$^{-1}$) of the particular gas species X and $D_X$ its diffusivity (i.e., the capability to allow diffusion, m$^2$ s$^{-1}$) in the studied vadose zone, and where $\frac{\mathrm{d}c_X}{\mathrm{d}z}$ describes the change of the concentration (mol m$^{-3}$) over depth (m). An alternative approach introduced by Soulet et al. (2018) uses gas accumulation chambers drilled into shallow weathering zones. Instead of calculating a carbon flux from a presupposed diffusion coefficient, which can introduce uncertainties (Maier & Schack-Kirchner, 2014), $CO_2$ release is measured in real-time in a similar way as commonly applied

to soil surfaces (Oertel et al., 2016). This method has provided new insight on how temperature, precipitation and topography control $CO_2$ emissions from marls (Soulet et al., 2021) and mudstones (Roylands et al., 2022). However, for weathering chambers that are installed within the rock face, three aspects remain unexplored: i) the rock volume that



contributes to the $CO_2$ accumulation measured in the chamber; ii) how the diffusive movement of $CO_2$ in the shallow weathering zone is impacted by short-term environmental changes (e.g., in temperature and hydrology); and iii) the quantification of $O_2$ depletion during oxidative weathering.

In this study, we investigate the weathering-driven exchange of $CO_2$ and $O_2$ by installing chambers into black shales and marls undergoing oxidation at two study sites in the steep terrains of the Draix-Bléone observatory, France (Draix-Bléone Observatory, 2015; Gaillardet et al., 2018; Klotz et al., 2023). Building on research from an outcrop at the same observatory (Soulet et al., 2021), we find that chamber-based $CO_2$ emissions vary significantly over one year, linked to changes in temperature and precipitation. A new theoretical framework is developed to refine $CO_2$ flux measurements (Sect. 4.1) and applied to quantify the rock pore space that is probed during a measurement (Sect. 4.2). This allows us to normalize $CO_2$ accumulation rates based on a contributing rock volume, and return estimates of fluxes emitted from the surface of rock outcrops. The resulting $CO_2$ fluxes can be accurately described as a function of temperature (Sect. 4.3). Using Fick's law, measurements of the $O_2$ concentration in the chambers are then used to quantify the $O_2$ consumption in the studied rocks (Sect. 4.4). Together, we provide new insights into the exchange of $CO_2$ and $O_2$ in the shallow weathering zone of sedimentary rocks.

## 2 Material and methods

### 2.1 Study area

The two study sites are located in the catchments of the Brusquet (area of 1.08 km$^2$) and Moulin (0.08 km$^2$) of the Draix-Bléone observatory (Draix-Bléone Observatory, 2015; Klotz et al., 2023), which is part of the French network of critical zone observatories (OZCAR) (Gaillardet et al., 2018). The Brusquet site is located at 44.16251° N 6.32330° E at 847 m.a.s.l. and the Moulin site at 44.14146° N 6.36095° E at 874 m.a.s.l. (Fig. 1). The catchments of the Draix-Bléone observatory have detailed measurements of river water discharge, river suspended load and bedload transport, and meteorological data over the last 4 decades (Carriere et al., 2020; Cras et al., 2007; Draix-Bléone Observatory, 2015; Klotz et al., 2023; Mallet et al., 2020; Mathys et al., 2003; Mathys & Klotz, 2008). Prior work has examined the occurrence of OC$_{petro}$ in the Brusquet, Moulin and Laval catchments (Copard et al., 2006; Graz et al., 2011, 2012). The Laval catchment (0.86 km$^2$), which neighbors the Moulin catchment (Fig. 1), is the location of previous in situ measurements of rock-derived $CO_2$ (Soulet et al., 2018, 2021).

The Moulin catchment overlays Callovian to Oxfordian marls (Graz et al., 2012; Janjou, 2004; Mathys et al., 2003). In contrast, the lithology of the Brusquet catchment consists of a sequence of Bajocian marly limestones, Aalenian marls and limestones to Toarcian black shales (Copard et al., 2006; Graz et al., 2011; Janjou, 2004), with the study site located on the latter (Fig. 1).

The climate is transitional between Alpine and Mediterranean with a hot and dry summer, including short and intense rainfall events during thunderstorms (up to 150 mm h$^{-1}$), with rainfall events of lower intensity typically during



spring and autumn (Carriere et al., 2020; Mallet et al., 2020; Mathys et al., 2003; Soulet et al., 2021). During winter, more than 100 days of frost can occur (Oostwoud Wijdenes & Ergenzinger, 1998; Rovéra & Robert, 2006) and frost-cracking from ice-segregation was found to control hillslope regolith production (Ariagno et al., 2022). The mean annual rainfall is ~ 900 mm and the mean annual air temperature is ~ 11 °C defined by high solar radiation (> 2,300 h yr$^{-1}$) (Mallet et al., 2020; Mathys & Klotz, 2008; Soulet et al., 2021).

Together, the climate and the erodible lithology of finely bedded, mechanically weak rocks result in a badland morphology with V-shaped gullies, high physical weathering rates and abrupt, sediment-loaded floods (Antoine et al., 1995; Cras et al., 2007; Le Bouteiller et al., 2021; Mathys et al., 2003). These features limit the development of soils and vegetation cover. In the late 19$^{th}$ century, following overgrazing in the wider area of the observatory, the Brusquet catchment was reforested (Cras et al., 2007; Mathys et al., 2003). Today ~ 87 % of the catchment area of Brusquet is vegetated, in contrast to ~ 46 % and ~ 32 % of the Moulin and Laval catchments, respectively (Carriere et al., 2020; Cras et al., 2007). The sediment export fluxes of the Brusquet catchment are on average ~ 70 t km$^{-2}$ yr$^{-1}$, and ~ 5,700 t km$^{-2}$ yr$^{-1}$ and ~ 14,300 t km$^{-2}$ yr$^{-1}$ for the Moulin and Laval catchments, respectively (Carriere et al., 2020; Mathys et al., 2003). Taking a regolith bulk density of ~ 1.3 t m$^{-3}$ - 1.8 t m$^{-3}$ into account (Ariagno et al., 2023; Bechet et al., 2016; Mallet et al., 2020; Mathys & Klotz, 2008; Oostwoud Wijdenes & Ergenzinger, 1998), a physical erosion rate of ~ 0.04 mm yr$^{-1}$ - 0.05 mm yr$^{-1}$, ~ 3.2 mm yr$^{-1}$ - 4.4 mm yr$^{-1}$ and ~ 8 mm yr$^{-1}$ - 11 mm yr$^{-1}$ can be estimated for the Brusquet, Moulin and Laval catchments, respectively. However, these values are catchment-scale averages, and the physical erosion can significantly vary spatially. On steep, bare slopes, the erosion rates may be more comparable between the catchments of the Draix-Bléone observatory (Bechet et al., 2016; Carriere et al., 2020; Mathys et al., 2003).

The bare surfaces in the catchments are characterized by four morphologically different layers: i) near surface, loose detrital cover of locally produced clasts or colluvial material with a thickness of ~ 0 - 10 cm; ii) below, the upper, fine regolith with a thickness of ~ 5 - 20 cm; and iii) the lower, coarse regolith with a thickness of ~ 10 - 20 cm; iv) the unweathered bedrock at the bottom (Maquaire et al., 2002; Mathys & Klotz, 2008; Oostwoud Wijdenes & Ergenzinger, 1998; Rovéra & Robert, 2006). The compactness and density of these layers increase, while the porosity decreases (from values of up to ~ 50 %), over depth towards the unweathered marl bedrock (grain density ~ 2.7 t m$^{-3}$, porosity ~ 10 - 20 %) (Bechet et al., 2016; Lofi et al., 2012; Mallet et al., 2020; Maquaire et al., 2002; Mathys & Klotz, 2008; Travelletti et al., 2012). The thickness of the weathering profile varies laterally with the thickest regolith layers and detrital cover on crests, minimal development in thalwegs and intermediate in gullies (Esteves et al., 2005; Maquaire et al., 2002).





**Figure 1: Location of the French Draix-Bléone observatory and of the study sites for in situ CO₂ and O₂ monitoring in the Brusquet catchment (red circle) and in the Moulin catchment (blue circle), alongside the location of previous research in the Laval catchment for reference (yellow circle)** (Soulet et al., 2018, 2021)**, and geological maps of these catchments** (Janjou, 2004)**. Meteorological stations are present at each of the catchment outlets with a maximum distance to the study sites of 200 m** (Draix-Bléone Observatory, 2015)**. Catchment-specific aerial imagery** (Draix-Bléone Observatory, 2015) **is shown alongside transparent aerial imagery of the wider area (2018 © IGN).**

## 2.2 Drilled gas accumulation chambers

To measure in situ the production of CO₂ and consumption of O₂ by oxidative weathering reactions, we use drilled chambers. The chambers were visited 6 times over the study to capture seasonal changes in weather conditions, on 27/09/2018, 11/01 - 14/01/2019, 11/04 - 15/04/2019, 27/05 - 29/05/2019, 05/07 - 12/07/2019 and 27/09 - 02/10/2019. Their design has been previously detailed (Soulet et al., 2018). In summary, a horizontal chamber is drilled directly into the exposed rock which has been cleared of detrital cover. The shape ensures a large surface to volume ratio to benefit measurement of gas concentrations and potential trapping of CO₂. To install the chambers of ~ 38 cm depth, a mechanical drill was used with a diameter of 2.9 cm. Rock powder left inside the chambers was blown away with a compressed-air gun.



A small PVC tube is inserted in the entrance of each chamber that is closed with a rubber stopper holding two glass tubes fitted with Tygon® tubing. The latter allow either connection to a gas-sampling system or sealing with WeLock® clips. To further isolate the chamber from the atmosphere, the intersection of the PVC tube and regolith is sealed with a silicone sealant (Unibond® Outdoor), which we previously tested to be free of potential contaminants for gas sampling (Roylands et al., 2022).

At both the Brusquet and Moulin study sites, we installed one array of 4 chambers placed in a square ($2 \times 2$) (Fig. 2) (Table 1). In each array, 2 chambers were placed in the same rock bed with a roughly horizontal orientation at the Brusquet site and roughly vertical at the Moulin site. The minimum distance between chambers was 70 cm. The aspect, hydrological and geomorphic setting of the location of both arrays is similar: they were placed at the upper margin of the watersheds in steep walls of gullies on a Southwest- (Brusquet) and South-facing aspect (Moulin). The chambers were drilled into bare rock faces devoid of roots and with minimal soil or vegetation cover in the vicinity to exclude a contribution by them to the $CO_2$ measurements (Fig. 2).

**Table 1: Characteristics of gas accumulation chambers drilled into weathering sedimentary rocks in the Brusquet catchment and in the Moulin catchment. For calculation of volume and inner surface area of the chambers, length and insertion depth of the PVC tube and rubber stopper are used.**

| Chamber identifier | | Site | Same bed as chamber | Installation date | Depth (cm) | Volume ($cm^3$) | Inner surface area ($cm^2$) |
|---|---|---|---|---|---|---|---|
| *short* | *long* | | | | | | |
| 1 | M-C-1 | Moulin | 3 | 24/09/2018 | 41.0 | 278 | 367 |
| 2 | M-A-2 | Moulin | 4 | 24/09/2018 | 39.5 | 263 | 346 |
| 3 | M-D-3 | Moulin | 1 | 24/09/2018 | 38.0 | 255 | 335 |
| 4 | M-B-4 | Moulin | 2 | 24/09/2018 | 39.0 | 262 | 345 |
| 5 | B-F-5 | Brusquet | 6 | 25/09/2018 | 38.0 | 255 | 335 |
| 6 | B-G-6 | Brusquet | 5 | 25/09/2018 | 37.0 | 255 | 335 |
| 7 | B-H-7 | Brusquet | 8 | 25/09/2018 | 35.0 | 229 | 299 |
| 8 | B-I-8 | Brusquet | 7 | 25/09/2018 | 35.0 | 235 | 308 |





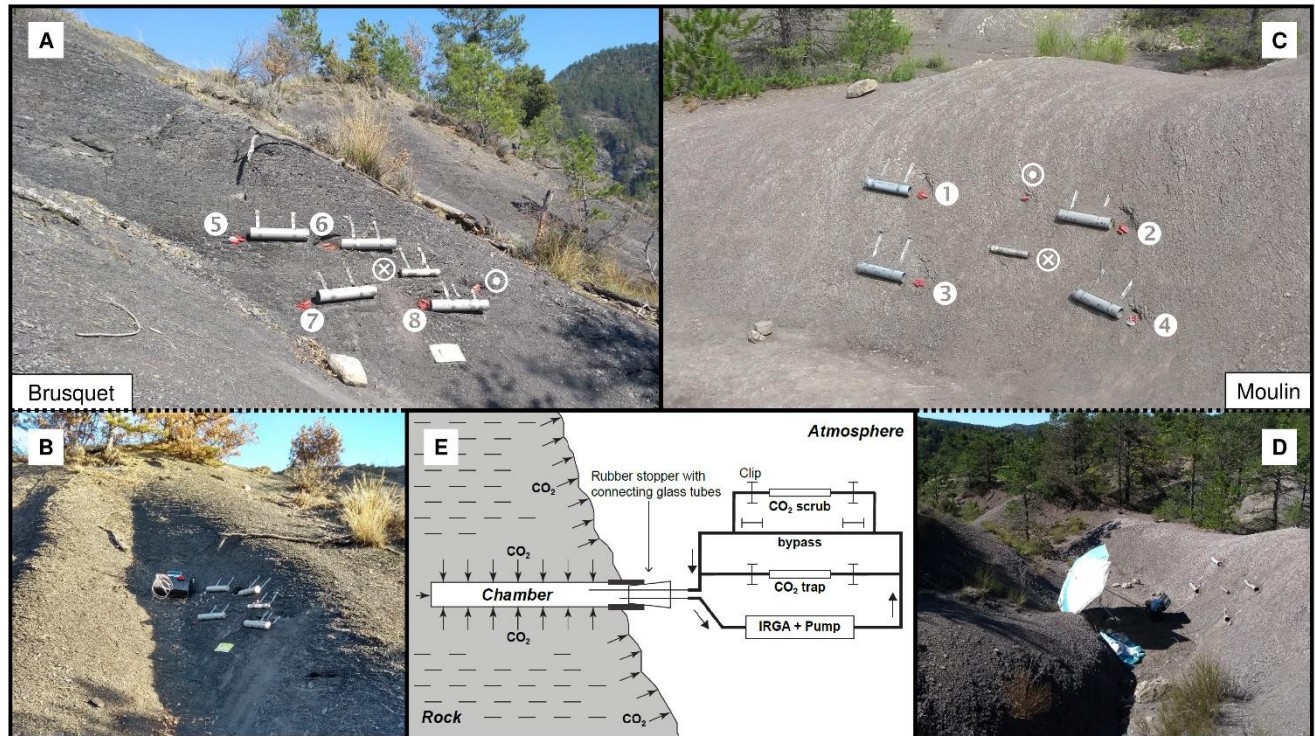

**Figure 2: The study sites in the Brusquet catchment (Panels A and B) and in the Moulin catchment (Panels C and D). Identifiers of the $CO_2$ accumulation chambers (Table 1) are given next to their entrance. Furthermore, the location of temperature and relative humidity loggers in a further chamber with similar properties (circled dot) and on the rock surface (circled X) are shown. For scale, the grey cases (not used in the present study) next to the chambers are of ~ 40 cm length. Panel E shows the design of the chambers and the sampling system adapted from** Soulet et al. (2018)**.**

### 2.3 Rock temperature and humidity

At both sites, a chamber with the same design was installed to hold a temperature and relative humidity logger (Lascar® EL-USB-2) (Fig. 2) from 27/09/2018 onwards. A second logger was placed on the rock surface (monitoring the air directly above it) with the main body fitted inside a small PVC tube for physical protection and an aluminum foil wrapping around the tube to avoid alteration of the temperature measurement due to the dark color of the housing of the sensor and the PVC tube.

Over the study period, technical issues with batteries of the temperature and relative humidity loggers prevented continuous data collection. To fill the gaps in the direct chamber temperature measurements, we use air temperatures from a local weather station as a proxy (Appendix A) by modifying a framework that describes soil temperatures by, amongst other variables, air temperature (Liang et al., 2014). Using the site-specific air temperatures, this approach simulates the chamber temperature well, with a root-mean-square error (RMSE) of 1.8 °C for the Brusquet catchment and 2.2 °C for the Moulin catchment (Appendix A).



## 2.4 Partial pressure of rock CO₂

The partial pressure of $CO_2$ ($pCO_2$) was measured alongside air pressure to determine the concentrations of $CO_2$ in the rock chambers with an infra-red gas analyzer (IRGA; EGM-5 Portable Gas Analyzer, PP Systems). This is equipped with an internal pump and calibrated to $pCO_2$ in the range of 0 ppmv to 30,000 ppmv. First, the closed-loop sampling system is purged of $CO_2$ using an inline $CO_2$ scrub (soda lime) (Hardie et al., 2005). This is then connected to a chamber to measure the ambient $pCO_2$. After a short equilibration, the $pCO_2$ in the chamber ($pCO_{2\ Chamber}$) is calculated from the $CO_2$ concentration in the combined air volume of the chamber and the sampling system by accounting for the dilution introduced from the $CO_2$-free air that was originally contained within the sampling system (Roylands et al., 2022; Soulet et al., 2018). To ensure that the determined $pCO_{2\ Chamber}$ is representative of the ambient $pCO_2$ in the rock pores around the chamber ($pCO_{2\ Rock}$), measurements are only considered if the chambers were left closed overnight so that the production of $CO_2$ from oxidative weathering could reach a steady-state in respect to diffusion between rock pores, chamber and atmosphere.

## 2.5 CO₂ flux measurements

Real-time measurements of $CO_2$ release in drilled chambers have been previously described in detail (Soulet et al., 2018) and used to quantify $CO_2$ flux (Roylands et al., 2022; Soulet et al., 2021). In summary, one $CO_2$ flux measurement consists of a series of repeated accumulations (typically 8 or more) that are recorded over time after determining the $pCO_{2\ Rock}$. First, the $pCO_{2\ Chamber}$ is lowered to a value close to the local atmosphere value using soda lime or a zeolite sieve (Sect. 2.6) (Fig. 2E). Then, $CO_2$ is allowed to build up, typically over ~ 6 min, before the $CO_2$ in the chamber is again removed to a near-atmospheric value. For each repeat, the rate of $CO_2$ accumulation $q$ (mgC min⁻¹) is calculated by fitting an exponential model to the recorded $pCO_2$ change, following Pirk et al. (2016):

$$\frac{dm(t)}{dt} = q - \lambda\left(m(t) - m_0\right),$$ (2)

where $\frac{dm(t)}{dt}$ is the carbon mass change (mgC) in the chamber with time (min), $m_0$ is the initial carbon mass in the chamber (mgC), and $\lambda$ (min⁻¹) is the curvature of the mass change that relates to diffusion of $CO_2$ between rock pores, chamber and atmosphere (Soulet et al., 2018). For this, the carbon mass ($m$, mgC) in the chamber is obtained from:

$$m = pCO_{2\ Chamber} \times V \times \frac{P}{R \times T} \times M_C \times 10^{-9},$$ (3)

where the measured $pCO_{2\ Chamber}$ is in ppmv (cm³ m⁻³), $V$ is the combined volume (cm³) of the chamber and the sampling system, $P$ is the pressure (Pa), $R$ is the universal gas constant (m³ Pa K⁻¹ mol⁻¹), $T$ is the chamber temperature (K), and $M_C$ is the molar mass of carbon (g mol⁻¹).

The $CO_2$ accumulation rate $q$ can be normalized to the internal surface area of the chamber $S_{Chamber}$ (i.e., area of exchange with the surrounding rock, m²) to account for differences in the depth of the chambers, which are related to differences in volume and surface area, giving a repeat-specific flux $Q$ (mgC min⁻¹ m⁻²):

$$Q = \frac{q}{S_{Chamber}}.$$ (4)



Alternatively, the $CO_2$ accumulation can be reported as a molar-based flux $J_{CO2}$ (mmol $CO_2$ min$^{-1}$ m$^{-2}$):

$$J_{CO2} = \frac{j_{CO2}}{S_{Chamber}} = \frac{q}{M_C \times S_{Chamber}} , \tag{5}$$

where $j_{CO2}$ (mmol $CO_2$ min$^{-1}$) is the molar-based analogue to $q$.

Previous work has noted that the $CO_2$ accumulation rate during the first measurements is typically higher than subsequent repeats (Soulet et al., 2018). To calculate a $CO_2$ flux from these repeated accumulations (consisting of $n$ repeats), previous work excluded the first 3 repeats ($q_1$ to $q_3$), and took the average of a minimum of 3 further repeats ($q_4$ to $q_{n \geq 6}$) (Roylands et al., 2022; Soulet et al., 2018, 2021). We examine this further using new data collected here, which also provides constraint on the nature of the gas exchange around the chambers.

## 2.6 $CO_2$ sampling

During a $CO_2$ flux measurement, the $CO_2$ in the chamber can be sampled by circulating it through a zeolite molecular sieve cartridge (MSC) mounted in parallel to the monitoring line (Hardie et al., 2005; Soulet et al., 2018). The volume of carbon loaded onto a sieve is estimated by adding up the $pCO_2$ maxima for each trapping episode minus the final $pCO_2$ after trapping (near-atmospheric value), while accounting for the combined volume of the chamber and the sampling system.

The zeolite sieves were heated in the laboratory to 425 °C and purged with high-purity nitrogen gas to release the $CO_2$ trapped onto them prior to cryogenic purification under vacuum (Garnett & Murray, 2013). The estimated sampled volume of $CO_2$ from the chamber-based $pCO_2$ measurements ($V_{CO2\text{-}IRGA}$, ml) can be compared with the volume recovered from the MSC in the laboratory ($V_{CO2\text{-}MSC}$, ml) giving a sampling ratio ($SR$, unitless) (Roylands et al., 2022):

$$SR = \frac{V_{CO2\text{-}MSC}}{V_{CO2\text{-}IRGA}} . \tag{6}$$

For this, all volumes of $CO_2$ are normalized to 0 °C and 1,013 mbar. The $SR$ thus allows us to independently check calculations of carbon mass using the $pCO_2$ data combined with the gas line and chamber volume measurements (Eq. 2 and 3).

## 2.7 Measuring $pO_2$ and $O_2$ fluxes

While measuring $pCO_2$, the EGM-5 Portable Gas Analyzer, incorporating the IRGA, also records the partial pressure of oxygen in the chamber ($pO_{2\ Chamber}$, % v/v) with an electrochemical $O_2$ sensor. The $pO_{2\ Chamber}$ cannot be used in the same way as the $pCO_2$ to quantify flux for two reasons: i) the precision of the $O_2$ sensor of $\geq 0.1$ % (v/v) is insufficient to observe real-time changes in $pO_{2\ Chamber}$; and ii) $O_2$ should be consumed during oxidative weathering and so we would require a method that replenishes oxygen, which was not done while measuring $CO_2$ accumulation.

        An alternative method to calculate an $O_2$ flux is based on Fick's law (Eq. 1), using the diffusive gradient of the partial pressure of $O_2$ in the rock ($pO_{2\ Rock}$) towards the atmospheric $O_2$ concentration ($pO_{2\ Atm.}$). To obtain $pO_{2\ Rock}$, the $pO_{2\ Chamber}$ measured after connecting to the chamber is corrected for the oxygen concentration in the sampling system. The $pO_2$ recorded before and during a measurement is corrected for instrument drift. The drift correction is based on measuring





the $p\mathrm{O}_{2\,\text{Atm.}}$ directly before or after a chamber-based measurement and assuming an average atmospheric oxygen concentration of 20.95 % (v/v).

To quantify the exchange of $\mathrm{O}_2$ between the chamber, connected rock pores and the atmosphere ($j_{\mathrm{O}2}$, mmol $\mathrm{O}_2$ min$^{-1}$), we describe the process via a diffusive transfer controlled by the diffusivity of $\mathrm{O}_2$ ($D_{\mathrm{O}2}$, cm$^2$ min$^{-1}$) across a spatial parameter $\omega$ (describing a combined movement over depth and area, cm$^1$ cm$^{-2}$):

$$j_{\mathrm{O}2} = \frac{D(\mathrm{O}2)}{\omega} \times (p\mathrm{O}_{2\,\text{Rock}} - p\mathrm{O}_{2\,\text{Atm.}}) \times \frac{P}{R \times T} \times 10^{-3} \ . \tag{7}$$

If we assume that $\omega$ is the same for $\mathrm{O}_2$ and $\mathrm{CO}_2$, linking the space of $\mathrm{O}_2$ consumption and $\mathrm{CO}_2$ release, $D_{\mathrm{O}2}$ can be
related to the diffusivity of $\mathrm{CO}_2$ ($D_{\mathrm{CO}2}$, cm$^2$ min$^{-1}$) based on their ideal relation in air, which is independent of temperature (Angert et al., 2015):

$$\frac{D(\mathrm{CO}2)}{D(\mathrm{O}2)} = 0.76 = \frac{D(\mathrm{CO}2)}{\omega} \div \frac{D(\mathrm{O}2)}{\omega} \ . \tag{8}$$

Differences in the effective diffusivities of gas species depend on the structure of the air-filled pore space, which is expected to have identical impacts on the gaseous movement of $\mathrm{O}_2$ and $\mathrm{CO}_2$ (Angert et al., 2015; Millington, 1959; Penman,
1940). Thus, if the term $\frac{D(\mathrm{CO}2)}{\omega}$ (cm$^3$ min$^{-1}$) can be quantified by other means (for instance, through analysis of the $p\mathrm{CO}_{2\,\text{Rock}}$ and $\mathrm{CO}_2$ flux data), we can quantify $j_{\mathrm{O}2}$. These themes will be discussed later (Sect. 4.1 and 4.4).

## 3 Results

### 3.1 Chamber temperature and meteorological conditions

Over the study period (27/09/2018 - 02/10/2019), similar variability in environmental conditions was recorded at the
Brusquet and Moulin catchment. The temperatures of the atmosphere, chamber interiors and at the rock surface showed daily and seasonal changes (Table 2) (Fig. 3A - F). Rainfall events were comparable in occurrence and extent, but the Brusquet catchment received less cumulative rainfall (773 mm) than the Moulin catchment (1033 mm) (Fig. 3G and 4H). The relative air humidity in the chambers was high and constant with values of ∼ 93.1 ± 4.5 % (± standard deviation, SD) and ∼ 91.3 ± 4.1 % at the Brusquet and Moulin sites, respectively (not considering gaps in the record) (Fig. 3I and 4J).




**Table 2: Overview of the variability of air temperature, chamber temperature and rock surface temperature over the study period (27/09/2018 - 02/10/2019) (Fig. 3). A gap in the record of rock surface temperatures at the Moulin site (25/10/2018 - 11/01/2019) is not considered.**

| Variable | Daily averages | | | Hourly resolution | |
|---|---|---|---|---|---|
| | Average (± SD) | Min. | Max. | Min. | Max. |
| Air temperature (°C) | | | | | |
| *Brusquet* | 10.5 ± 7.4 | -4.9 | 29.5 | -9.5 | 41.5 |
| *Moulin* | 10.5 ± 7.3 | -4.0 | 27.9 | -9.1 | 38.3 |
| Chamber temperature (°C) | | | | | |
| *Brusquet* | 15.7 ± 9.4 | 0.7 | 34.6 | -1.1 | 41.0 |
| *Moulin* | 16.6 ± 9.7 | 0.2 | 33.0 | -1.5 | 36.1 |
| Rock surface temperature (°C) | | | | | |
| *Brusquet* | 14.4 ± 8.7 | -1.6 | 35.3 | -7.0 | 56.0 |
| *Moulin* | 17.9 ± 9.0 | -1.4 | 37.3 | -8.5 | 62.5 |









**Figure 3: Environmental variables for weathering chambers in 2018 and 2019, with grey shaded areas showing fieldwork visits, in the Brusquet catchment (red) and in the Moulin catchment (blue). Daily averages are shown by darker colors in all panels. Panels A and B: hourly air temperatures from meteorological stations** (Draix-Bléone Observatory, 2015)**. Panels C and D: hourly chamber temperatures. Estimated chamber temperatures are indicated by lighter colors (Sect. 2.3) and are shown for gaps in the logger record (denoted). Panels E and F: hourly rock surface temperatures. Panels G and H: daily rainfall. Panels I and J: relative** 265 **humidity in the chambers with gaps in the record similar to the chamber temperatures.**

### 3.2 $p\mathrm{CO_2}$ measurements and $\mathrm{CO_2}$ collection

The $p\mathrm{CO_{2\,Rock}}$ values varied between the chambers and over time. In the Brusquet catchment, the observed $p\mathrm{CO_{2\,Rock}}$ values were on average 1,490 ± 743 ppmv (± SD, if not reported otherwise, n = 28), and 1,492 ± 633 (n = 32) in the Moulin catchment (Table 3).

**Table 3: Chamber-specific overview of $p\mathrm{CO_{2\,Rock}}$, with variations over time reported as 1 SD.**

| Chamber identifiers | | Site | $p\mathrm{CO_{2\,Rock}}$ (ppmv) | | | |
|---|---|---|---|---|---|---|
| *short* | *long* | | Average | n | Min. | Max. |
| 5 | B-F-5 | Brusquet | 861 ± 254 | 5 | 588 | 1,167 |
| 6 | B-G-6 | Brusquet | 1,985 ± 771 | 12 | 936 | 3,378 |
| 7 | B-H-7 | Brusquet | 936 ± 340 | 4 | 588 | 1,399 |
| 8 | B-I-8 | Brusquet | 1,405 ± 438 | 7 | 721 | 2,000 |
| *Brusquet totals* | | | *1,490 ± 743* | *28* | | |
| 1 | M-C-1 | Moulin | 1,740 ± 654 | 10 | 551 | 2,499 |
| 2 | M-A-2 | Moulin | 720 ± 110 | 4 | 543 | 834 |
| 3 | M-D-3 | Moulin | 1,881 ± 127 | 3 | 1,755 | 2,054 |
| 4 | M-B-4 | Moulin | 1,456 ± 576 | 15 | 681 | 2,680 |
| *Moulin totals* | | | *1,492 ± 633* | *32* | | |

Following the determination of $p\mathrm{CO_{2\,Rock}}$, a total of 37 $\mathrm{CO_2}$ flux measurements were conducted in the Brusquet catchment, of which 32 consisted of ≥ 8 repeats. In the Moulin catchment, 41 measurements were made, with 37 having ≥ 8 repeats. Every individual $\mathrm{CO_2}$ flux measurement showed an initial decline of accumulation rates that approached a constant value of peak $\mathrm{CO_2}$ concentration (Fig. 4). Considering the repeats 6 - 8, averages of the $\mathrm{CO_2}$ accumulation rates varied 275 between chambers and over time at each single chamber, with occurrence of the lowest accumulation rates in winter and highest in summer. On four visits, a chamber was measured twice a day and the observed $\mathrm{CO_2}$ release was higher in the afternoon than in the morning, coinciding with an increase of the chamber temperature. Overall, the observed $\mathrm{CO_2}$ accumulation rates (averages of $q_6$ to $q_8$) were on average 15.2 ± 11.7 µgC min⁻¹ (n = 32) and 11.5 ± 8.0 µgC min⁻¹ (n = 37) in the Brusquet catchment and in the Moulin catchment, respectively. The associated values of the fitting parameter $\lambda$ (Eq. 2) 280 were on average of 0.179 ± 0.076 min⁻¹ (n = 32) and 0.140 ± 0.061 min⁻¹ (n = 37) in the Brusquet catchment and in the Moulin catchment, respectively.

At each study site, $\mathrm{CO_2}$ was sampled from two chambers. Following recovery from the zeolite sieves in the laboratory, the sampling ratio (*SR*, Eq. 6) was quantified with an overall median of 1.03 (Table 4).



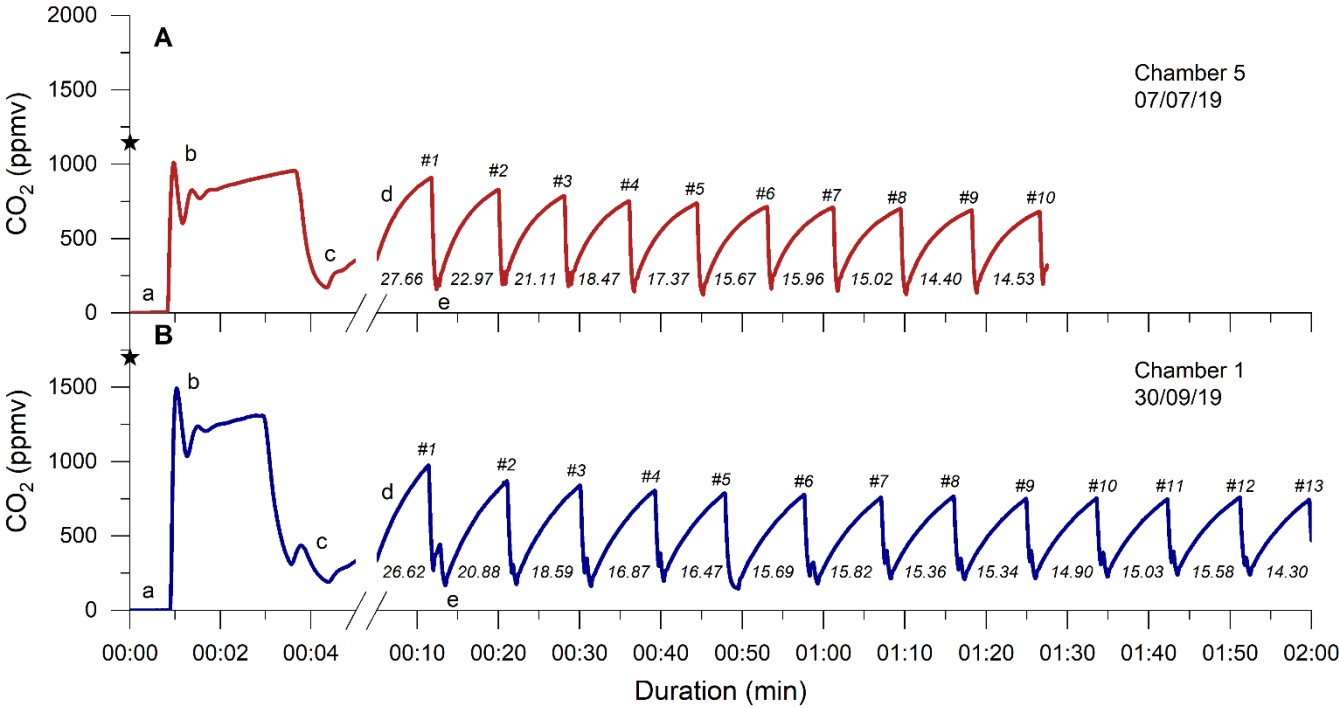

**Figure 4: Two examples of monitoring the CO₂ concentration (ppmv) in a chamber during a flux measurement. Following connection, the CO₂-free air of the sampling system (a) equilibrates with the partial pressure of CO₂ in the chamber (b), which is representative of $p$CO₂ Rock (denoted by ★). After the CO₂ in the chamber is removed to a near-atmospheric value (sometimes stepwise, c), the first accumulation (d) is monitored (with a change in axis scale), followed by further removal (e) and accumulation events (~ 6 min, numbers denoted with #). The measured accumulation rates ($q$, mgC min⁻¹ per chamber) are given for each repeat. The CO₂ flux measurement of chamber 5 on the 07/07/19 (Brusquet catchment, Panel A, red) consisted of 10 repeats, and the measurement of chamber 1 on the 30/09/19 (Moulin catchment, Panel B, blue) consisted of 13 repeats.**

**Table 4: Overview of the chamber-specific sampling ratio ($SR$, Eq. 6), which compares the estimated volumes of CO₂ sampled in the Brusquet catchment and in the Moulin catchment with volumes recovered in the laboratory from zeolite sieves.**

| Chamber identifiers | | Site | Number | Sampling ratio ($SR$, unitless) | | | |
|---|---|---|---|---|---|---|---|
| *short* | *long* | | of samples | Median | Average (± 1 SD) | | |
| 1 | M-C-1 | Moulin | 4 | 0.90 | 0.98 | ± | 0.25 |
| 4 | M-B-4 | Moulin | 7 | 0.93 | 0.94 | ± | 0.10 |
| 6 | B-G-6 | Brusquet | 7 | 1.04 | 1.05 | ± | 0.09 |
| 8 | B-I-8 | Brusquet | 2 | 1.12 | 1.12 | ± | 0.04 |
| *Totals* | | | *20* | *1.03* | *1.00* | *±* | *0.15* |

### 3.3 $p$O₂ measurements

The total number of usable $p$O₂ Chamber measurements was limited to 15. The difference in $p$O₂ between the chambers and the atmosphere ($p$O₂ Chamber - $p$O₂ Atm.) varied over time, ranging from zero within uncertainty (0.18 ± 0.36 %, v/v, ± 95 %





confidence interval, CI) to $-1.50 \pm 0.30$ % (v/v), with the lowest $p$O$_{2\,Chamber}$ values in summer and higher values (lower gradient) in winter at both sites.

## 4 Discussion

Carbon dioxide release during oxidative weathering of sedimentary rocks exposed in steep mountain areas has been shown to vary with changes in temperature, precipitation and local topography (Roylands et al., 2022; Soulet et al., 2021). Furthermore, previous studies on weathering profiles (Bolton et al., 2006; Petsch, 2014) and on the chemical composition of rivers (Bufe et al., 2021; Calmels et al., 2007; Hilton et al., 2021) indicated that geomorphological and hydrological factors are important controls on the release of CO$_2$ and the consumption of O$_2$ during oxidative weathering. The fluxes should also

depend on the pore space characteristics of the weathering zone, such as porosity and tortuosity (Bolton et al., 2006; Brantley et al., 2013; Gu et al., 2020a, 2020b; Soulet et al., 2021). However, we lack direct observations of how the chemical and physical properties of the weathering zone affect the in situ fluxes of CO$_2$ and O$_2$. In addition, only with information on the contributing rock volume to a measured rock-derived flux can we upscale and quantify CO$_2$ and O$_2$ fluxes from the measurement site to the landscape scale.

In the following discussion, we first propose a new approach of interpreting in situ CO$_2$ flux measurements (Sect. 4.1.1) that allows us to assess the diffusion of CO$_2$ and O$_2$ in the shallow critical zone (Sect. 4.1.2). This can be used to quantify the rock volume contributing to the measured fluxes (Data-flow diagram in Appendix B) (Sect. 4.2). We then examine the implications of these new insights for quantifying the rock-derived CO$_2$ release (Sect. 4.3), and then determine the coinciding O$_2$ consumption (Appendix B) to investigate an overall redox budget of oxidative weathering in an erosive

environment (Sect. 4.4).

### 4.1 Probing the gas exchange of the shallow critical zone

### 4.1.1 Explaining the patterns of CO$_2$ accumulation during a single flux measurement

To explain the initial decline of CO$_2$ accumulation rates during a flux measurement that stabilizes over time (Fig. 4), we consider the known volume of a drilled chamber and distinguish it from the unknown rock pore space around it. After

arriving at a chamber to start a CO$_2$ flux measurement, the initial scrubbing (before the first repeat $q_1$) removes CO$_2$ from the chamber (i.e., $p$CO$_{2\,Chamber}$) to a near-atmospheric level, but we assume that it does not remove the CO$_2$ from the connected pore space to a similar $p$CO$_2$ value. Repeated scrubbing and removal of CO$_2$ after CO$_2$ accumulations (Fig. 4) then acts to lower the CO$_2$ concentration in the rock pore space connected to the chamber ($p$CO$_{2\,Rock}$). Once this pool of "excess" CO$_2$ has been exhausted, subsequent CO$_2$ accumulation rates reach a plateau and are assumed to represent the real-time

production and diffusion of CO$_2$ in the rock surrounding the chamber. This explanation requires that the air volume processed by the sampling system equals the chamber volume, whereas the gas exchange between chamber and connected rock pores happens solely via diffusion. This assumption is supported by the measured CO$_2$ sampling ratio, $SR$ (Eq. 6), with





an average $SR = 1.03 \pm 0.15$ (Table 4), showing we effectively trap the chamber contents. This is similar to the recovery efficiency of ~ 95 % of $CO_2$ standards in the laboratory (Garnett et al., 2019).

To interpret the period where we consider an excess of $CO_2$ is diffusing into the chamber from a connected pore space, we introduce an exponential fitting model that describes the decrease of $CO_2$ accumulation rates ($q$) over time (Fig. 5):

$$q(t) = \alpha \times \exp(-\beta \times t) + q_{Plateau} , \tag{9}$$

where $q_{Plateau}$ is a constant value of the plateaued $CO_2$ accumulation rate, the sum of $\alpha$ and $q_{Plateau}$ is the initial rate of accumulation ($\approx q_1$, mgC min$^{-1}$) at the start of the flux measurement (t = 0), $\beta$ is the measurement-specific removal constant

(min$^{-1}$), and the term $\alpha \times \exp(-\beta \times t)$ describes the purging of the initially stored $pCO_2$ from the rock pore space connected to the chamber over time.

To ensure reliable results from fitting the exponential model, measurements are only considered if the last 3 of at least 8 repeats ($q_{n-2}$ to $q_n$ for n ≥ 8) are "stabilized", which we define as having a relative standard deviation of less than 5 %. To interpret the remaining $CO_2$ flux measurements, chamber-specific averages of the removal constant $\beta$ from stabilized

measurements are used to extrapolate $q_{Plateau}$ for measurements that did not stabilize.

The outputs of this analysis provide a $CO_2$ flux that represents the real-time production of $CO_2$ ($q_{Plateau}$), while quantifying the scrubbing of $CO_2$ stored initially in the connected volume of pore space around each chamber. This allows us to assess the diffusive movement of $CO_2$ in the shallow weathering zone (Sect. 4.1.2) and to estimate the contributing rock volume (Sect. 4.2) (Appendix B).

Earth **Surface**
**Dynamics**
Discussions



**Figure 5: Examples of CO$_2$ flux measurements consisting of several repeated accumulation rate measurements. Measured accumulation rates ($q$, µgC min$^{-1}$ per chamber, y-axis) are shown alongside exponential fits (Eq. 9) describing their evolution over time (x-axis) and alongside the 95 % confidence intervals of the modeled level at which the rates plateau ($q_{Plateau}$) for chambers in the Brusquet catchment (Panel A) and in the Moulin catchment (Panel B). Dates and fitting parameters α (µgC min$^{-1}$) and β (min$^{-1}$) of the single flux measurements are denoted. For a comparison of the stabilizing evolution of different CO$_2$ flux measurements, indicated by varying colors, $q$(t) is normalized to $q_{Plateau}$ for both study sites in separate plots (Panel C).**

### 4.1.2 Assessing the diffusivity of the shallow weathering zone

After a CO$_2$ flux measurement, the chamber is re-sealed. The chamber interior and surrounding pore space will evolve to a steady-state of diffusive movement of CO$_2$ along a concentration gradient between the surface of the rock outcrop and the atmosphere so that $p$CO$_2$ $_{Chamber}$ = $p$CO$_2$ $_{Rock}$. Thus, this steady-state of a closed chamber differs from the manipulated environment of a CO$_2$ flux measurement (Fig. 4). The comparison of the two states can shed light on gas movement and the





physical properties of the rocks undergoing weathering. Here, we explore how the observed changes in $p\mathrm{CO}_{2\,\mathrm{Rock}}$ and $\mathrm{CO}_2$ fluxes can be explained by a framework of diffusive processes in the shallow critical zone, and assess the degree to which these are modulated by weather conditions.

According to Fick's law (Eq. 1), diffusion of gases in a porous medium is controlled by: i) the production and accumulation of $\mathrm{CO}_2$; ii) the volume of space (rock pores and/or chamber) and length scale over which molecules travel towards the low-$p\mathrm{CO}_2$ reservoir; and iii) the diffusivity of $\mathrm{CO}_2$ along their path. We find a co-variation of $p\mathrm{CO}_{2\,\mathrm{Rock}}$ and the $\mathrm{CO}_2$ fluxes that is similar for both sites, a relationship that can be explained by a linear regression model (Appendix C), with high $p\mathrm{CO}_{2\,\mathrm{Rock}}$ values coinciding with high $\mathrm{CO}_2$ accumulation rates (Fig. 6A). This indicates that the contributing volume of
rock pores and the diffusivity (the remaining variables from Fick's law) may be stable at both sites over the study period.

        However, the ambient hydroclimate appears to modify the response of these variables. We consider measurements as being made during "wet" or "dry" periods, whereby "wet" measurements are those where the cumulative precipitation over the last 3 days was $\geq 5$ mm. At a given $p\mathrm{CO}_{2\,\mathrm{Rock}}$ value, "dry" conditions are associated with lower $\mathrm{CO}_2$ production compared to "wet" conditions (Fig. 6A). However, previous research has shown that rock-derived $\mathrm{CO}_2$ fluxes from drilled
chambers are lower following rain events, but recover subsequently over a few dry days (Roylands et al., 2022; Soulet et al., 2021). This observation has been linked to the degree of water saturation controlling the gas motion of $\mathrm{O}_2$ and $\mathrm{CO}_2$, as well as to dissolution of weathering derived carbon and subsequent export of dissolved inorganic carbon (Roylands et al., 2022; Soulet et al., 2021). It is important to note that a decrease of the production of $\mathrm{CO}_2$, associated with a lower $\mathrm{O}_2$ supply required for the oxidative weathering reactions and/or with a greater uptake of carbon into the DIC, would also decrease
$p\mathrm{CO}_{2\,\mathrm{Rock}}$ (Roylands et al., 2022; Soulet et al., 2021). Thus, a change in the relationship between $p\mathrm{CO}_{2\,\mathrm{Rock}}$ and $\mathrm{CO}_2$ flux may be better explained by differences in the diffusivity or the contributing rock volume during "wet" and "dry" conditions.

        According to Fick's law, a lower diffusivity at a constant contributing volume of rock results in higher $p\mathrm{CO}_{2\,\mathrm{Rock}}$ values. Thus, "wet" conditions may be associated with a decrease in the diffusivity of gases in the weathering rocks. This fits a simple model describing the effective diffusivity $D_{\mathrm{Rock}}$ ($\mathrm{m^2\ s^{-1}}$) of a given gas in porous media (such as rocks and soils) at a
given temperature by:

$$D_{\mathrm{Rock}} = D_{\mathrm{Air}} \times \varphi_{\mathrm{Air\text{-}filled}} \times \tau \ , \tag{10}$$

where $D_{\mathrm{Air}}$ is the diffusion coefficient ($\mathrm{m^2\ s^{-1}}$) of the particular gas in air, $\tau$ is a dimensionless tortuosity factor, and $\varphi_{\mathrm{Air\text{-}filled}}$ is the air-filled porosity (v/v %) (Davidson & Trumbore, 1995; Penman, 1940). If $\varphi_{\mathrm{Air\text{-}filled}}$ decreases due to meteoric water filling the pore space, precipitation events are likely to lower the effective diffusivity of $\mathrm{CO}_2$ within the critical zone. An
increase of moisture in porous media also leads to more tortuous pathways (Davidson & Trumbore, 1995; Millington, 1959), which could further lower $D_{\mathrm{Rock}}$ under wet conditions. Analogously, rock moisture would also affect the diffusion of atmospheric $\mathrm{O}_2$ into the rock pore space, so that this framework can explain the observed decrease of $\mathrm{CO}_2$ production following rain events (Roylands et al., 2022; Soulet et al., 2021).





To describe the diffusion of $CO_2$ during the steady-state of a closed chamber, we use Fick's law (Eq. 1) and the measured $CO_2$ flux ($q_{Plateau}$) and the concentration gradient of $CO_2$ ($pCO_{2\,Rock}$ - $pCO_{2\,Atm.}$) to define a measure ($\frac{D(CO2)}{\omega}$, cm³ min⁻¹) that describes the effective diffusivity $D_{CO2}$ (cm² min⁻¹) of the $CO_2$ flux towards the atmosphere over the unknown effective depth and area $\omega$ (cm¹ cm⁻²):

$$\frac{D(CO2)}{\omega} = \frac{q_{Plateau}}{pCO_{2\,Rock} - pCO_{2\,Atm.}} \times \frac{R \times T}{P} \times \frac{10^9}{M_C}.$$ (11)

The calculated values (based on repeats 6 - 8) are on average $27.5 \pm 12.4$ cm³ min⁻¹ (n = 25) and $21.8 \pm 13.2$ cm³ min⁻¹ (n = 30) for the Brusquet catchment and the Moulin catchment, respectively.

An alternative way to assess diffusivity is to use the constant $\lambda$ (Eq. 2) describing the curvature of the repeated accumulations during a $CO_2$ flux measurement (Fig. 4) (Pirk et al., 2016). Differences between $\lambda$ and $\frac{D(CO2)}{\omega}$ may be expected because $\lambda$ is representative of short intervals (~ 6 min observations), while $\frac{D(CO2)}{\omega}$ represents a period of a few hours. We find a significant linear correlation of $\lambda$ and $\frac{D(CO2)}{\omega}$ for all samples irrespective of the study site (Fig. 6B) (Appendix D). The similarities of both metrics affirm that the accumulation rates determined during flux measurements are representative for the longer-term $CO_2$ release towards the atmosphere.

The concordance of changes in $\lambda$ and in $\frac{D(CO2)}{\omega}$ suggests that the rock pore space is relatively homogenous in porosity and tortuosity, since the diffusive pathways of the steady-state during a stabilized flux measurement differ from that of a closed chamber (Appendix E). Minor heterogeneities may explain some scatter in the correlation of $\lambda$ and $\frac{D(CO2)}{\omega}$, as well as short-term changes in the effective rock space contributing $CO_2$ to a chamber induced by percolation of meteoric waters.

In more detail, we find some variability in the measures of diffusivity linked to the hydroclimatic conditions. "Wet" conditions coincide generally with somewhat lower $\frac{D(CO2)}{\omega}$ values for a given $\lambda$ (Fig. 6B) (Appendix D). Because the atmosphere is acting as the low-$pCO_2$ reservoir during the steady-state of a closed chamber, $\frac{D(CO2)}{\omega}$ is likely to be more influenced by surficial processes than $\lambda$, which is affected by $CO_2$ migration pathways towards the chamber (Appendix E). For example, lower $\frac{D(CO2)}{\omega}$ values for a given $\lambda$ may be the result of filling of surficial cracks with water, or micro-landslides, swelling of the surface rock material and lateral expansion following rainfall events (Bechet et al., 2015), which may hinder the migration of gas. During drier conditions, cracks may significantly increase gas exchange between the rocks and the atmosphere (Maier & Schack-Kirchner, 2014; Weisbrod et al., 2009). In the study area, desiccation cracks typically appear at steep slopes during summer, when erosion by runoff is less prevailing than in spring and autumn, whereas a thick layer of loose detrital cover can be accumulated during winter due to frost weathering (Ariagno et al., 2022, 2023), when movement of surface materials is limited to solifluction (Bechet et al., 2016). Thus, the diffusivity of the rock surface presumably changes over time, with greater values during dry summer conditions (Fig. 6F and 7I).



The $\lambda$ and $\frac{D(CO2)}{\omega}$ values can also be explored as a function of temperature inside the chambers (Fig. 6F and 7I). In air, the diffusion coefficient of $CO_2$ is strongly controlled by temperature with an increase by a factor of ~ 1.25 at 35 °C

compared to 0 °C (Massman, 1998). However, we find a much larger change of $\lambda$ and $\frac{D(CO2)}{\omega}$, with an average increase by a factor of ~ 3.5 between 0 °C and 35 °C (Fig. 6F and 7I). This relation between temperature and diffusivity could be explained by a coinciding decrease of rock moisture. In the marls of the Laval catchment, neighboring the studied Moulin catchment, lower near-surface water contents were observed during dry summer periods, with values as low as ~ 10 % contrasting to values of up to ~ 25 % in winter (Mallet et al., 2020). However, the relation of rock moisture and temperature

is not straightforward (Soulet et al., 2021), with precipitation being an important control on surface rock moisture. In addition, we observe high and constant relative air humidity in the chambers over the year (Fig. 3). Together, a complex hydrological control on $D_{Rock}$, which includes surface processes, may explain some part of the high apparent temperature sensitivity of $\lambda$ and $\frac{D(CO2)}{\omega}$ by modifying $\tau$ and $\varphi_{Air\text{-filled}}$ (Eq. 10), alongside changes of $D_{Air}$ forced solely by temperature.

In summary, disentangling diffusive processes in the shallow weathering zone is complicated by drivers that can be

interrelated and co-vary (Fig. 6). This is also commonly observed in soils (Davidson & Trumbore, 1995; Hashimoto & Komatsu, 2006; Maier & Schack-Kirchner, 2014; Tokunaga et al., 2016). Generally, hydrology and temperature are important controls on $p CO_{2\ Rock}$, $CO_2$ flux, diffusivity and potentially rock pore space, all of which contribute to the release of $CO_2$ to the atmosphere. Interestingly, similar responses to changes in environmental controls are observed at both study sites, and they appear to have similar diffusivity measures. However, the $CO_2$ fluxes differ significantly between sites, with

greater $CO_2$ efflux at a given rock temperature from the chambers in the Brusquet catchment compared to the Moulin catchment (Fig. 6C), which may be explained by a difference in the source of $CO_2$ or by differences in the contributing rock volume (Sect. 4.2).

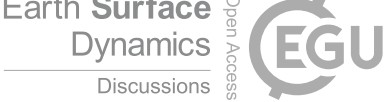





**Figure 6: Comparisons of the diffusivity measures $\lambda$ and $\frac{D(CO2)}{\omega}$, CO₂ accumulation rates, $p$CO₂ Rock values and chamber**
**temperatures. Color coding differentiates "dry" (orange) from "wet" samples (violet) using a threshold of a cumulative**
**precipitation of 5 mm over the last 3 days. The origin of samples is indicated with open circles for the Brusquet catchment and**
**filled triangles for the Moulin catchment. For consistency, all parameters determined during CO₂ flux measurements are**
**calculated on the basis of the repeats 6 - 8 (error bars: 1 SD). Estimated temperatures are indicated by accompanying error bars**
**(RMSE).**

**4.2 Assessing the contributing rock pore volume**

**4.2.1 Quantification of the contributing rock pore volume**

Chamber-based measurements of CO₂ flux provide insight on the variability of fluxes over time and the environmental
controls that force them (e.g., Bond-Lamberty & Thomson, 2010; Oertel et al., 2016; Pirk et al., 2016; Roylands et al., 2022;
Soulet et al., 2021). However, the volume of material that contributes to the measured CO₂ fluxes is rarely quantified. If this
could be determined, the production of CO₂ can be considered in terms of the mass of reactants, allowing comparisons
between different field sites and laboratory experiments (e.g., Angert et al., 2015; Kalks et al., 2021; Lefèvre et al., 2014;
Soucémarianadin et al., 2018; Tokunaga et al., 2016). In the case of the internal rock chambers used here, quantification of
the contributing rock volume would allow us to upscale the fluxes over an outcrop surface area. To do this, we use the
exponential fitting model (Eq. 9) that describes the transition between a closed chamber and the manipulated state during
flux measurements (Fig. 4 and 5) as a way to quantify the carbon mass derived from the rock pore space. By doing so, we
can use the $p$CO₂ Rock to calculate the corresponding air volume in the rock volume contributing CO₂. The volume of rock
pores, in turn, is used to estimate the corresponding rock volume and its geometry, and, ultimately, the rock mass to
determine an absolute weathering flux.

        First the mass of CO₂ purged during a flux measurement from the rock pore space around the chamber is described
as an excess of CO₂ (CO₂ Excess, mgC):

$$CO_{2\,Excess} = \int_{t(0)}^{t(Plateau)} \alpha \times \exp(-\beta \times t) \,,\tag{12}$$

with $\alpha$ and $\beta$ being the fitting parameters from the same fitting procedure used to calculate $q_{Plateau}$ (Eq. 9) over time, starting
at the beginning of the flux measurement (t = 0) and ending when the integrated term approaches zero ($t_{Plateau}$, when $q(t)$
equals $q_{Plateau}$). The air volume of the rock pores can be estimated from CO₂ Excess by using the $p$CO₂ Rock at the start of the flux
measurement (when $p$CO₂ Rock equals $p$CO₂ Chamber). This air volume ($V_{Rock\,pores}$, cm³) is calculated by modifying Eq. 3:

$$V_{Rock\,pores} = CO_{2\,Excess} \times \frac{R \times T}{P} \times \frac{10^9}{M_C \times p CO_{2\,Rock}} \,.\tag{13}$$

        Overall, the calculated values of $V_{Rock\,pores}$ are similar for both study sites with $365 \pm 208$ cm³ ($\pm$ 1 SD of the average
of measurement-specific values) for the chambers in the Brusquet catchment, and $322 \pm 174$ cm³ for the chambers in the
Moulin catchment (Fig. 7) (Table 5). However, significant variation is observed over time for each chamber (Fig. 8), while
each measurement-specific value of $V_{Rock\,pores}$ is associated with a high uncertainty (Table 5). These uncertainties are not
normally distributed, with an average upper relative uncertainty of $125.8 \pm 140.1$ % (average of 95 % CI $\pm$ 1 SD) and an
average lower relative uncertainty of $47.8 \pm 42.3$ % for all samples.



Earth **Surface**
**Dynamics**
Discussions



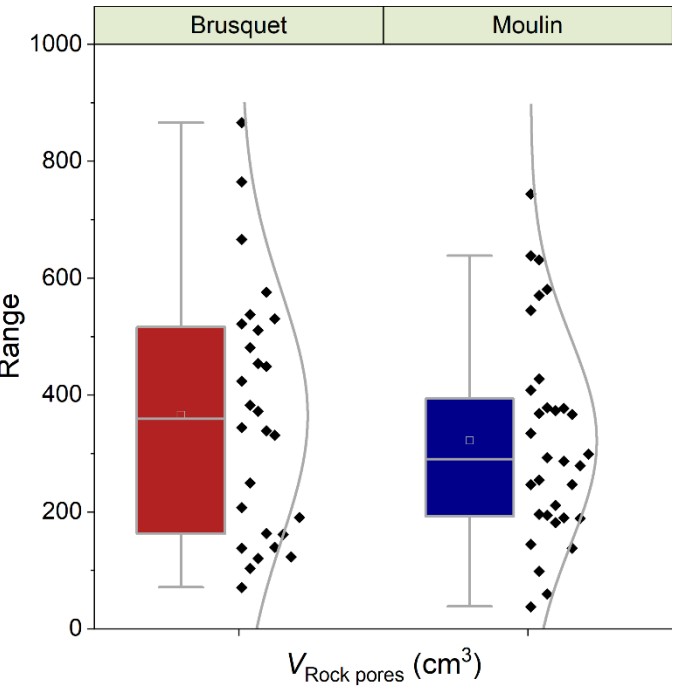

**Figure 7: Catchment-specific box plots and distribution curves summarizing the volume of rock pores ($V_{\text{Rock pores}}$) connected to**
**each chamber determined during $CO_2$ flux measurements including 4 chambers at each site. Boxes indicate the 25 % - 75 % range**
**alongside the 1.5 interquartile ranges with mean (open square) and median (line). Colors indicate the origin (Brusquet: red,**
**Moulin: blue).**

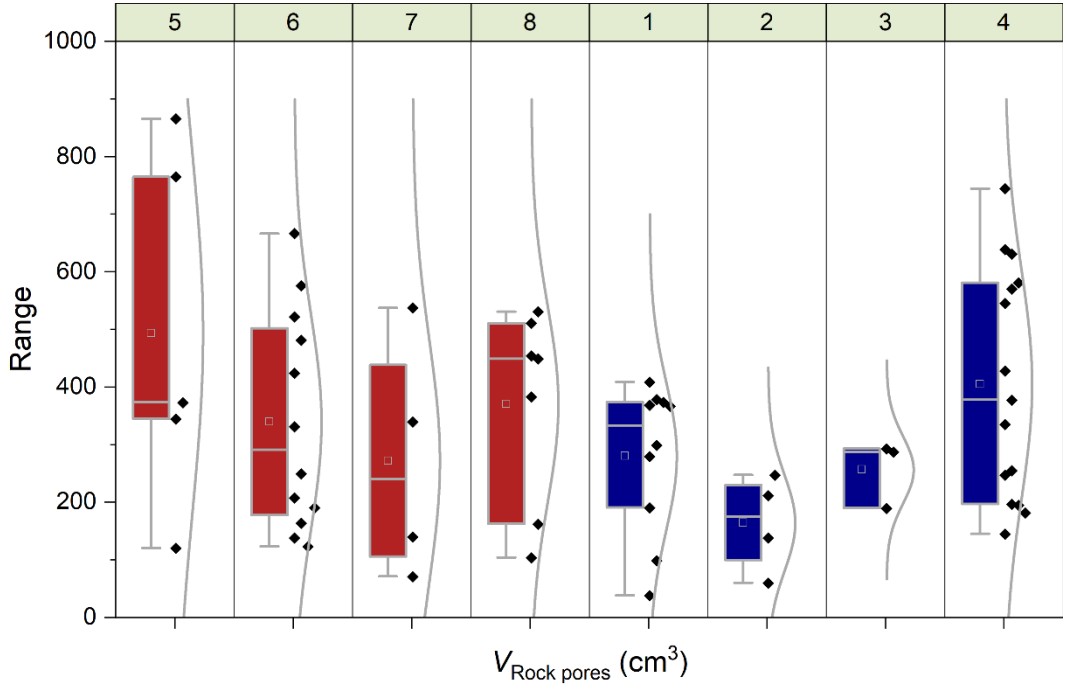





**Figure 8: Chamber-specific box plots and distribution curves summarizing the volume of rock pores ($V_{\text{Rock pores}}$) connected to each chamber determined during $CO_2$ flux measurements. Boxes indicate the 25 % - 75 % range alongside the 1.5 interquartile ranges with mean (open square) and median (line). Panels at the top show the chamber identifiers (Table 1) and colors indicate the origin (Brusquet: red, sites 5 - 8; Moulin: blue, sites 1 - 4).**

### 4.2.2 Environmental controls on the contributing rock pore volume

The variation of $V_{\text{Rock pores}}$ may be linked to changes in the diffusive processes and weather conditions (Fig. 9). Overall, higher values of $\lambda$ coincide with greater values of $V_{\text{Rock pores}}$ (all samples: $R^2$ of a linear regression = 0.52, p = <0.001, n = 55) (Fig. 9A). This is also true for the relation between $V_{\text{Rock pores}}$ and $\frac{D(CO2)}{\omega}$ (Fig. 9B), which itself is positively correlated to $\lambda$ (Sect. 4.1). These relationships are similar for both sites, and for "wet" and "dry" conditions. The latter observation indicates that rock moisture impacts the diffusivity of $CO_2$ and $V_{\text{Rock pores}}$ in equal measure. This, in turn, is in line with Fick's law, with the extent of the rock pore space that contributes $CO_2$ to a chamber depending on the potential of gas to move within the rocks undergoing weathering. In this process, the degree to which changes in the diffusivity impact the contributing rock volume is driven by the effective change in length of the diffusion paths. Here, a change of $\lambda$ from 0.1 min$^{-1}$ to 0.2 min$^{-1}$ is associated with a change of $V_{\text{Rock pores}}$ by a factor of ~ 3.5 and this roughly fits the modification of the geometry of a cylinder-shaped rock pore space around a drilled chamber when doubling its effective radius.

Furthermore, differences in $V_{\text{Rock pores}}$ coincide with changes in temperature (all samples: $R^2$ of a linear regression = 0.47, p = <0.001, n = 60) (Fig. 9C). This coincidence is important to consider when assessing the control of temperature on the $CO_2$ production from chemical weathering (i.e., weathering kinetics), and is discussed later (Sect. 4.3). The coincidence of positive correlations of temperature and $CO_2$ production, and of temperature and the extent of $V_{\text{Rock pores}}$ (indirectly connected by $D_{CO2}$) also means that changes in $CO_2$ flux are associated with changes in the contributing rock pore space (all samples: $R^2$ of a linear regression = 0.42, p = <0.001, n = 60) (Fig. 9D).





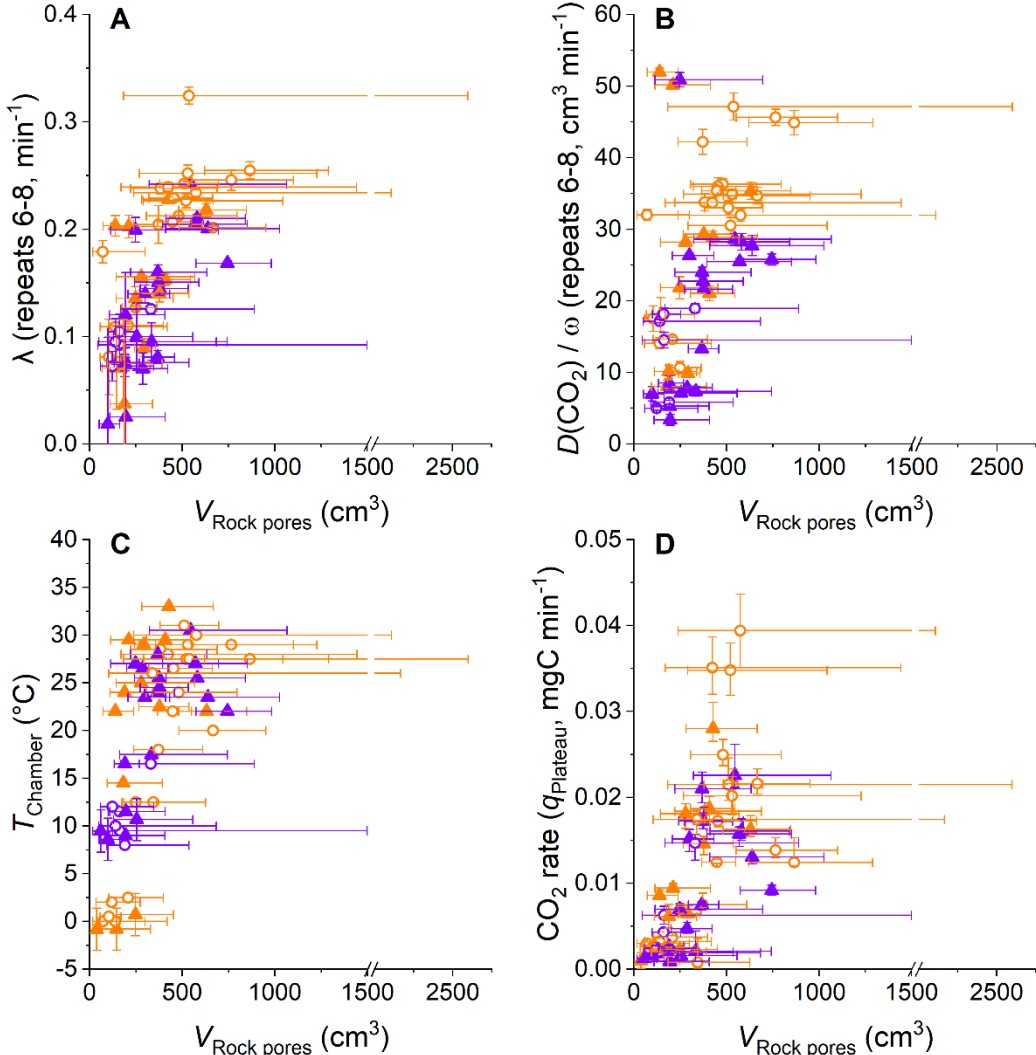

**Figure 9: Comparisons of $V_{Rock\ pores}$, the diffusivity measures $\lambda$ and $\frac{D(CO2)}{\omega}$, CO₂ accumulation rates, and chamber temperatures. Color coding differentiates "dry" (orange) from "wet" samples (violet) using a threshold of a cumulative precipitation of 5 mm over the last 3 days. The origin of samples is indicated with open circles for the Brusquet catchment and filled triangles for the Moulin catchment. Reported values of $\lambda$ and $\frac{D(CO2)}{\omega}$, are based on the repeats 6 - 8 of a CO₂ flux measurement (error bars: 1 SD), whereas the calculation of $V_{Rock\ pores}$ and CO₂ accumulation rates are based on the fitting model (error bars: 95 % CI). Estimated temperatures are indicated by accompanying error bars (RMSE).**

### 4.2.3 Upscaling chamber-based CO₂ fluxes

The determined $V_{Rock\ pores}$ can be combined with the porosity of the rocks undergoing weathering to quantify the volume of rock contributing CO₂ to flux measurement ($V_{Rock}$, cm³):

$$V_{Rock} = \frac{V_{Rock\ pores}}{\varphi_{Air\text{-}filled}}. \tag{14}$$





This assumes that the majority of rock pores are well connected (total porosity ≈ connected porosity), with no significant water filling of the pore space. An effective air-filled porosity of 30 %, within a 95 % confidence interval of 20 % - 40 %, is used based on porosity and water saturation measurements from the Draix-Bléone observatory (Garel et al., 2012; Mallet et al., 2020).

On average, $CO_2$ fluxes from chambers in the Brusquet catchment derive from a rock volume of $1,216.0^{+3,081.0}_{-690.2}$ cm³ (within 95 % confidence interval based on propagating the uncertainties of the fitting procedure and of the assigned porosity), which is similar to the release of $CO_2$ in the Moulin catchment from rock volumes of $1,071.8^{+1,606.1}_{-555.4}$ cm³. If we visualize this volume as a cylindrical rock layer around the drilled chambers (Fig. 2E: sampling distance indicated by arrows pointing towards the chamber), its thickness would be ~ $1.9^{+2.1}_{-0.9}$ cm. However, the geometry of this space is unknown.

Instead, considering that porosities are highest at the surface of rock outcrops in the study area (Lofi et al., 2012; Mallet et al., 2020; Maquaire et al., 2002; Mathys & Klotz, 2008; Travelletti et al., 2012), where unloading and climatic controls on physical weathering act most efficiently (Ariagno et al., 2022; Bechet et al., 2015, 2016; Cras et al., 2007; Mathys & Klotz, 2008), the shape of the porous and permeable rock that contributes to gas exchange is likely to be more like a cone around a chamber with a radius that declines over depth.

The knowledge of the probed layer thickness can be combined with the inner surface area of the chambers to give the spatial parameter $\omega$ (Eq. 11) and to calculate the effective diffusivity of $CO_2$ in the air-filled rock pores zone. Overall, we find values of $D_{CO2}$ ranging between ~ 0.02 cm² min⁻¹ and ~ 0.30 cm² min⁻¹ (considering the range of $\frac{D(CO2)}{\omega}$ at both study sites; Sect. 4.1). Interestingly, these values are similar to diffusion coefficients that were determined by laboratory experiments at 22.5 °C, with $O_2$ as the tracer gas, which correspond to $D_{CO2}$ values of 0.34 cm² min⁻¹ and 0.43 cm² min⁻¹ for

limestones with porosities of 40 % and 46 %, and of 0.04 cm² min⁻¹ and 0.17 cm² min⁻¹ for mudstones with porosities of 33 % and 38 %, respectively (Peng et al., 2012).

The rock volume around a chamber can be "unwrapped" to assess a surface area on an outcrop that would have the same contributing rock pore volume ($S_{Rock}$, cm²). This can be done if the weathering depth $z_{Rock}$ (cm) over that $CO_2$ is thought to be produced by oxidative weathering is considered:

$$S_{Rock} = \frac{V_{Rock}}{z_{Rock}}.$$    (15)

The $z_{Rock}$ value can be inferred from the morphology of bare surfaces in the study area (Maquaire et al., 2002; Mathys & Klotz, 2008; Oostwoud Wijdenes & Ergenzinger, 1998; Rovéra & Robert, 2006), based on the assumption that chemical weathering of sedimentary rocks occurs at the same depths where physical properties are altered (Brantley et al., 2013; Gu et al., 2020a, 2020b; Lebedeva & Brantley, 2020). Accordingly, based on previous research in a neighboring catchment reporting physical alteration that extends to depths of ~ $20.0^{+10.0}_{-10.0}$ cm at slopes similar to our study sites (Maquaire et al., 2002), we estimate the chemical weathering to extend to similar depths at both study sites.

The corresponding values calculated for $S_{Rock}$ can be compared to the inner surface area of the chambers (Table 1). On average, $S_{Rock}$ is smaller than the inner surface area of the drilled chambers, by a factor of $6.4^{+16.1}_{-5.3}$. This means that $CO_2$





fluxes from chambers drilled into rocks and normalized to the chamber inner surface area (Eq. 4) cannot be compared

directly with topographic surface fluxes (e.g., from surface chambers), which are typically reported for soils (Bond-Lamberty & Thomson, 2010; Oertel et al., 2016). Instead, $CO_2$ fluxes from a drilled chamber need to be corrected by considering $V_{Rock}$ and the weathering depth. On average, the measured $CO_2$ fluxes (Table 5) equate to a topographic surface efflux of ~ 1,215 tC $km^{-2}$ $yr^{-1}$ and of ~ 885 tC $km^{-2}$ $yr^{-1}$ in the Brusquet catchment and in the Moulin catchment, respectively, which is similar to the global mean emissions of $CO_2$ from soils with different land cover (Oertel et al., 2016).

Carbon fluxes from oxidative weathering can be linked to a rock mass, allowing fluxes to be interpreted in terms of the overall kinetics of the oxidative weathering reactions. This is essential for theoretical carbon cycle modeling (Bao et al., 2017; Bolton et al., 2006). Following calculation of $V_{Rock}$, the rock mass emitting $CO_2$ ($m_{Rock}$, g) can be estimated by using an average estimate of the density of the rock grains surrounding the chambers ($\rho_{Rock}$, g $cm^{-3}$ = t $m^{-3}$):

$$m_{Rock} = (V_{Rock} - V_{Rock\ pores}) \times \rho_{Rock} . \tag{16}$$

Considering a grain density of $2.7^{+0.02}_{-0.02}$ t $m^{-3}$ (Lofi et al., 2012), we find that an average rock mass of $2,153.0^{+5,318.4}_{-1,315.5}$ g produces the $CO_2$ fluxes derived from chambers in the Brusquet catchment and in the Moulin catchment. To our knowledge, this allows for the first time an absolute report of weathering-derived $CO_2$ fluxes that are measured in real-time and in situ.

The combined quantification of $CO_2$ fluxes and of the corresponding rock mass undergoing oxidative weathering

means that there is an opportunity for future research to include investigations of the internal surface area of the studied rocks, which would allow reporting field-based $CO_2$ fluxes normalized to the reacting surface areas. Such normalizations are typically considered during modeling (Bao et al., 2017; Bolton et al., 2006) to acknowledge that the internal surface area can change significantly during sedimentary rock weathering (Fischer & Gaupp, 2005). Analogously to silicate weathering rates, variations in $OC_{petro}$ and carbonate weathering rates obtained from different field and laboratory conditions may be related to

the conceptualization and parametrization of the reactive surface area, which needs to be considered when comparing them (Brantley et al., 2007; White & Brantley, 2003).





**Table 5: Chamber-specific overview of CO₂ Excess, $V_{Rock\ pores}$ and CO₂ accumulation rate ($q_{Plateau}$), including catchment-specific summaries. Uncertainties of minima and maxima are representing the 95 % CI, whereas averages are reported with 1 SD.**

| Chamber identifiers | | Site | CO$_2$ Excess (µgC) | | | | $V_{Rock\ pores}$ (cm$^3$) | | | |
|---|---|---|---|---|---|---|---|---|---|---|
| *short* | *long* | | Average | n | Min. | Max. | Average | n | Min. | Max. |
| 5 | B-F-5 | Brusquet | 222 ± 166 | 5 | $34^{+44}_{-24}$ | $444^{+78}_{-44}$ | 493 ± 279 | 5 | $120^{+153}_{-85}$ | $866^{+424}_{-244}$ |
| 6 | B-G-6 | Brusquet | 353 ± 267 | 12 | $64^{+130}_{-37}$ | $863^{+1,709}_{-505}$ | 339 ± 180 | 12 | $123^{+223}_{-64}$ | $666^{+286}_{-183}$ |
| 7 | B-H-7 | Brusquet | 142 ± 126 | 4 | $20^{+66}_{-16}$ | $336^{+1,352}_{-222}$ | 272 ± 182 | 4 | $71^{+229}_{-54}$ | $537^{+2159}_{-354}$ |
| 8 | B-I-8 | Brusquet | 261 ± 147 | 7 | $37^{+24}_{-16}$ | $450^{+164}_{-115}$ | 370 ± 157 | 7 | $103^{+67}_{-46}$ | $530^{+697}_{-262}$ |
| *Brusquet totals* | | | *277 ± 221* | *28* | | | *365 ± 208* | *28* | | |
| 1 | M-C-1 | Moulin | 252 ± 150 | 10 | $10^{+57}_{-10}$ | $453^{+152}_{-113}$ | 280 ± 122 | 10 | $38^{+210}_{-38}$ | $408^{+137}_{-102}$ |
| 2 | M-A-2 | Moulin | 55 ± 26 | 4 | $15^{+17}_{-11}$ | $79^{+144}_{-42}$ | 164 ± 72 | 4 | $60^{+65}_{-42}$ | $247^{+449}_{-133}$ |
| 3 | M-D-3 | Moulin | 217 ± 50 | 3 | $149^{+119}_{-62}$ | $267^{+59}_{-46}$ | 256 ± 48 | 3 | $189^{+150}_{-79}$ | $293^{+64}_{-51}$ |
| 4 | M-B-4 | Moulin | 296 ± 192 | 15 | $48^{+62}_{-26}$ | $569^{+544}_{-232}$ | 404 ± 193 | 15 | $145^{+184}_{-77}$ | $744^{+238}_{-169}$ |
| *Moulin totals* | | | *245 ± 174* | *32* | | | *322 ± 174* | *32* | | |

| Chamber identifiers | | Site | CO$_2$ rate (µgC min$^{-1}$) | | | |
|---|---|---|---|---|---|---|
| *short* | *long* | | Average | n | Min. | Max. |
| 5 | B-F-5 | Brusquet | 6.6 ± 5.9 | 10 | $0.8^{+2.8}_{-0.7}$ | $18.5^{+5.0}_{-1.5}$ |
| 6 | B-G-6 | Brusquet | 17.1 ± 13.5 | 13 | $2.0^{+0.2}_{-0.4}$ | $39.4^{+4.3}_{-4.3}$ |
| 7 | B-H-7 | Brusquet | 13.8 ± 9.6 | 5 | $1.9^{+0.1}_{-0.3}$ | $25.4^{+3.3}_{-2.4}$ |
| 8 | B-I-8 | Brusquet | 14.7 ± 7.0 | 8 | $2.6^{+0.3}_{-0.3}$ | $21.5^{+3.1}_{-1.8}$ |
| *Brusquet totals* | | | *13.2 ± 10.8* | *36* | | |
| 1 | M-C-1 | Moulin | 12.5 ± 7.4 | 11 | $1.0^{+0.9}_{-0.9}$ | $21.0^{+1.9}_{-1.6}$ |
| 2 | M-A-2 | Moulin | 5.0 ± 3.5 | 6 | $1.2^{+0.3}_{-0.3}$ | $9.4^{+0.5}_{-0.4}$ |
| 3 | M-D-3 | Moulin | 4.6 ± 3.1 | 6 | $0.5^{+0.4}_{-0.4}$ | $9.3^{+1.2}_{-0.3}$ |
| 4 | M-B-4 | Moulin | 11.1 ± 8.4 | 18 | $0.9^{+0.1}_{-0.3}$ | $28.0^{+3.0}_{-1.5}$ |
| *Moulin totals* | | | *9.6 ± 7.7* | *41* | | |

### 570  4.3 Implications for CO₂ flux measurements

#### 4.3.1 Accuracy of CO₂ flux measurements

The time-dependency of carbon accumulations during the CO₂ flux measurements (Fig. 4 and 5) has provided new insights into the nature of the shallow weathering zone (Sect. 4.1 and 4.2) and has important implications for how CO₂ fluxes are





quantified. In short, changes of $pCO_{2\ Chamber}$ during the field operations are interpreted as the combination of: i) purging of

$CO_2$ stored initially in the chamber and surrounding rock pores ($pCO_{2\ Rock}$); and ii) the real-time production of $CO_2$ from oxidative weathering. At the start of a $CO_2$ flux measurement, $CO_2$ accumulations have important contributions from i), which led previous studies to use the later repeats ($q_4$ to $q_{n \geq 6}$) to quantify ii) (Roylands et al., 2022; Soulet et al., 2018, 2021). Here, longer measurements at each chamber allow us to explore this in more detail.

     The time it takes for the $CO_2$ accumulation rates to decline and stabilize within the 95 % confidence interval of the

$q_{Plateau}$ value derived from fitting $CO_2$ accumulation rates over time (Eq. 9) is ~ 90 min corresponding to ~ 10 repeats (Fig. 5). This implies that treating only the first 3 repeats of a measurement as combined signals of purging and production of $CO_2$, as done previously in similar studies (Roylands et al., 2022; Soulet et al., 2018, 2021), returns a greater flux than $q_{Plateau}$. However, the overestimate is modest: the average accumulation rate of the measured repeats 4 - 6 is ~ 15 % higher than the $q_{Plateau}$ value (Fig. 10A). This is true for the entire data set, including stabilized and extrapolated measurements, and for the

site-specific samples. The relative offset is constant irrespective of the overall size of the $CO_2$ accumulation (Fig. 10A), which means that flux data including non-plateaued accumulations can be corrected. It also means that any link between $CO_2$ flux and measured environmental variables is robust (Roylands et al., 2022; Soulet et al., 2021). When taking the average of the measured repeats 6 - 8 (Fig. 10B), the value is ~ 7 % higher than $q_{Plateau}$.

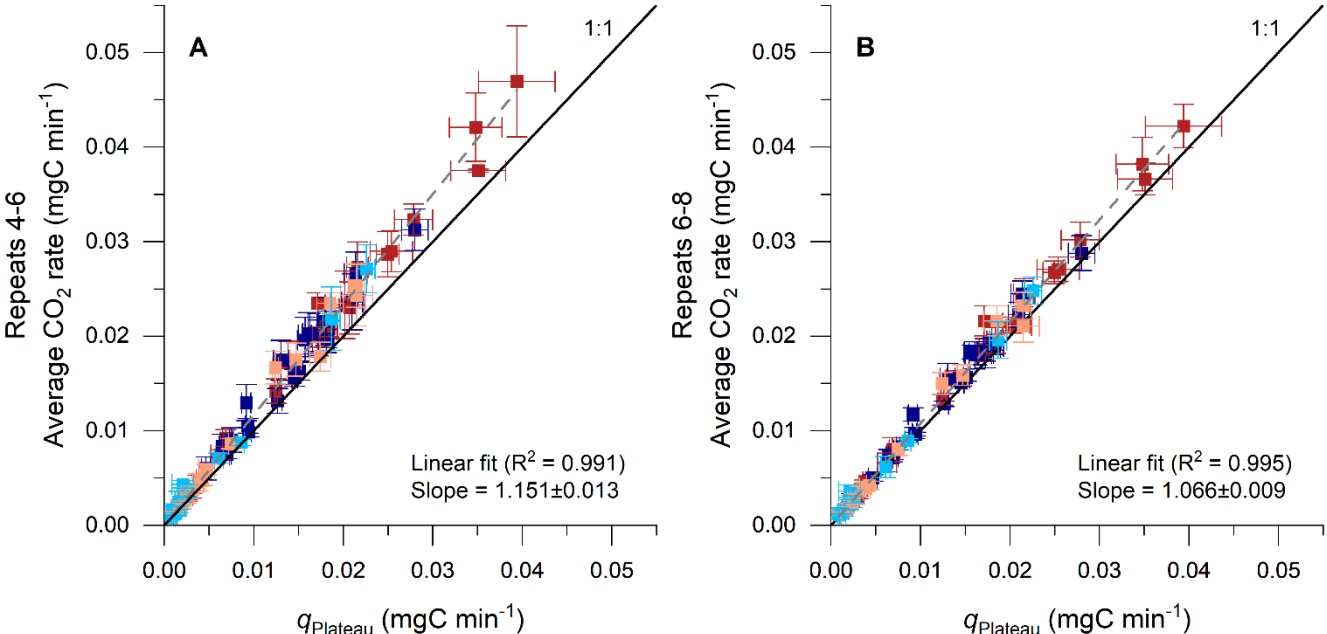

**Figure 10: Comparison of CO₂ fluxes determined by using the average of repeated accumulation rates and by an exponential fitting model. Results from the proposed model are given on the x-axis ($q_{Plateau}$, Eq. 9; error bars: 95 % CI) and are compared with the averages of the repeats 4 - 6 (Panel A) and of the repeats 6 - 8 (Panel B) on the y-axis (error bars: 2 SD), alongside a linear regression and a 1:1 line for reference. Red colors indicate samples from the Brusquet catchment (stabilized: dark, extrapolated fitting: light), and blue colors indicate samples from the Moulin catchment (stabilized: dark, extrapolated fitting: light).**



### 4.3.2 Reporting CO$_2$ flux as a function of temperature

Recent research has highlighted that temperature controls the release of CO$_2$ from chambers drilled into the shallow weathering zone of sedimentary rocks (Soulet et al., 2021), with an exponential response:

$$F = F_0 \times \exp(\gamma\, T) , \tag{17}$$

where $F$ is the CO$_2$ flux (mgC m$^{-2}$ d$^{-1}$, using the chamber-specific inner surface area), $T$ is the temperature (°C), $F_0$ is the CO$_2$ flux at 0 °C and $\gamma$ is the growth rate parameter (°C$^{-1}$) that is derived from an exponential model. For the oxidation of marls, Soulet et al. (2021) found a value for $\gamma$ of $0.070 \pm 0.007$ °C$^{-1}$ ($\pm$ standard error) considering five different chambers independent of their $F_0$ values, based on daily-averaged chamber temperatures. Across differences in the hydrological setting of these chambers, $F_0$ values ranged from $35 \pm 7$ mgC m$^{-2}$ d$^{-1}$ to $626 \pm 113$ mgC m$^{-2}$ d$^{-1}$ with the lowest CO$_2$ fluxes in close proximity to a riverbed (Soulet et al., 2021).

Here, at different installation sites, we find a similar exponential response of CO$_2$ release to temperature with $\gamma$ values of $0.065 \pm 0.012$ °C$^{-1}$ (Brusquet) and $0.067 \pm 0.018$ °C$^{-1}$ (Moulin). However, using an hourly resolution for the chamber temperature (Fig. 11A) returns higher $\gamma$ values ($0.077 \pm 0.013$ °C$^{-1}$ at Brusquet; $0.085 \pm 0.016$ °C$^{-1}$ at Moulin). This increase in $\gamma$ can be explained by an instantaneous response of weathering reactions to in situ temperature changes, and fits to the observation that CO$_2$ fluxes increased over a few hours alongside increases of chamber temperature for chambers visited twice a day (Appendix F). Overall, this observation highlights the importance of considering the in situ environmental conditions with a high temporal resolution (Sect. 3.1).

Changes in temperature also coincide with changes in the diffusive processes in the rocks surrounding a chamber (Sect. 4.1). To differentiate changes in the diffusive framework from the CO$_2$ production at a given temperature (i.e., weathering kinetics), the CO$_2$ fluxes can be normalized to $V_{\text{Rock pores}}$, which is representative for the contributing amount of rock grains undergoing oxidation (Sect. 4.2). Since the observed CO$_2$ fluxes range by a factor of $\sim 18.2$, while values of $V_{\text{Rock pores}}$ exhibit a lower range of a factor of $\sim 5.9$ (Table 5), this normalization does not diminish the importance of the temperature control on the CO$_2$ release (Fig. 11C). However, due to large measurement-specific uncertainties that are associated with the calculation of $V_{\text{Rock pores}}$, a full assessment of whether higher chamber-derived CO$_2$ fluxes at higher temperatures are partly a result of greater contributing rock volumes is hindered.

Despite the similarities of the topography, hydrology, erosion rates (Fig. 1 and 2) and $V_{\text{Rock pores}}$ (Fig. 7 and 8), we find site-specific differences in the bulk CO$_2$ production at a given temperature, which may be linked to differences in the source of carbon associated with the different rock types outcropping in the Brusquet catchment (black shales; $F_0 = 122.2 \pm 41.3$ mgC m$^{-2}$ d$^{-1}$) and in the Moulin catchment (marls; $F_0 = 45.6 \pm 20.3$ mgC m$^{-2}$ d$^{-1}$). To better understand these different weathering fluxes, future research is needed to assess the carbon source(s) and the response of the weathering reactions to changes in temperature at both study sites in more detail, for instance, by studying the chemical composition of the rocks (i.e., contents of OC$_{\text{petro}}$, carbonates and sulfides) alongside the radiocarbon and stable carbon isotope composition



Earth **Surface**
**Dynamics**
Discussions



of the released $CO_2$, analogously to previous research in the neighboring Laval catchment (Soulet et al., 2018, 2021) and in the Waiapu catchment in New Zealand (Roylands et al., 2022).

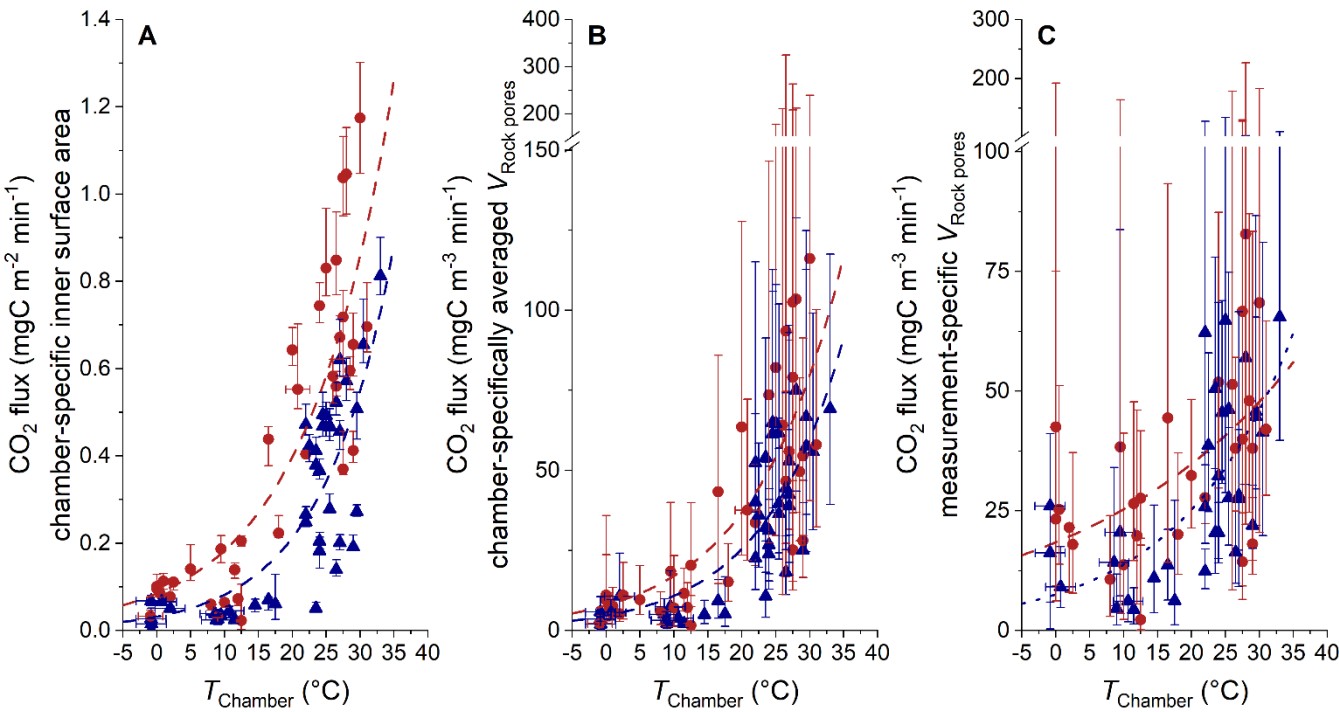

**Figure 11: Comparison of $CO_2$ fluxes with different normalizations and chamber temperature. Panel A: normalization to the chamber-specific inner surface area (Table 1) (including the 95 % CI of $q_{Plateau}$). Panel B: normalization to chamber-specifically averaged $V_{Rock\ pores}$ (Table 5) (including the combined uncertainty of $q_{Plateau}$ and the averaged 95 % CI of $V_{Rock\ pores}$). Panel C: normalization to measurement-specific $V_{Rock\ pores}$ (including the combined uncertainty of $q_{Plateau}$ and the measurement-specific 95 % CI of $V_{Rock\ pores}$). Estimated temperatures are indicated by accompanying error bars (RMSE). Colors indicate the site of measurement (Brusquet: red circles, Moulin: blue triangles).**

### 4.4 Linking $CO_2$ and $O_2$ fluxes

The $O_2$ consumption in the weathering zone provides a tool for investigating the kinetics of sedimentary rock weathering (Tune et al., 2020, 2023), analogous to previous research on soils (Angert et al., 2015; Hicks Pries et al., 2020). Overall, $pO_{2\ Chamber}$ values in the Brusquet catchment and in the Moulin catchment were similar or lower than the $O_2$ concentration of the atmosphere, confirming that the weathering zones are sinks of oxygen. The variation of $pO_{2\ Chamber}$ over time coincides with a variation in temperature within the chambers (Fig. 12A), with roughly similar relationships for both study sites. This observation fits to the recently proposed importance of temperature controlling oxidative weathering kinetics (Soulet et al., 2021). However, due to a limited number of samples and a large measurement uncertainty, neither chamber-specific differences nor the impact of precipitation on $pO_{2\ Chamber}$ can be evaluated accurately.

The observed $pO_{2\ Chamber}$ values can be used to calculate a diffusive flux of $O_2$ between the chamber, connected rock pores and the atmosphere (Eq. 7 and 8). This is based on the insights into the diffusive processes of the chambers and the



connected rock space that come from the $CO_2$ measurements (Sect. 2.7 and 4.1) (Appendix B). Previous work on porous media has established that the effective diffusivities of $CO_2$ and $O_2$ are impacted in a similar way by the structure of the air-filled pore space (Angert et al., 2015; Millington, 1959; Penman, 1940), which allows determination of the effective
diffusivity of $O_2$ from that of $CO_2$ in the shallow weathering zone.

The calculated $O_2$ fluxes are representative of the same rock volume that is releasing $CO_2$, which also means that a report of absolute fluxes is possible (Sect. 4.2). Altogether, the chamber-specific $O_2$ exchange rate in the Brusquet catchment and the Moulin catchment range between zero within uncertainty ($0.42^{+0.58}_{-0.88}$ µmol $O_2$ min$^{-1}$) to a maximum consumption of $O_2$ of $-16.33^{+5.72}_{-7.75}$ µmol $O_2$ min$^{-1}$, with increasing $O_2$ consumption with increasing temperature (Fig. 12B). The $O_2$ fluxes
have a greater relative uncertainty compared to the $pO_2$ gradient because they include the measurement-specific diffusivity.

The $O_2$ flux into the chambers and their connected rock pores can be compared with the $CO_2$ flux from this space. At 20 °C, an $O_2$ consumption rate of $\sim -8.7$ µmol $O_2$ min$^{-1}$ coincides with an average $CO_2$ accumulation rate of $\sim 1.1$ µmol $CO_2$ min$^{-1}$ in the Brusquet catchment and of $\sim 0.6$ µmol $CO_2$ min$^{-1}$ in the Moulin catchment. This is an average ratio of $\sim 1$ mol $O_2 : 0.1$ mol $CO_2$. This field-based molar ratio of $O_2$ consumption and $CO_2$ release is significantly lower than
the theoretical ratio of weathering reactions describing the oxidation of sedimentary rocks. For example, the oxidation of $OC_{petro}$ is theoretically described by a ratio of 1 mol $O_2 : 1$ mol $CO_2$ (Petsch, 2014). In addition, the oxidation of pyrite minerals coupled to the dissolution of carbonates is theoretically characterized by a ratio of up to 1.875 mol $O_2 : 1$ mol $CO_2$ if the $CO_2$ release occurs in situ (Soulet et al., 2021; Torres et al., 2014). To investigate this discrepancy, here we discuss several mechanisms that could influence the consumption of $O_2$ and the release of $CO_2$ in the shallow weathering zone.

In addition to $OC_{petro}$ and pyrite minerals, other minerals, such as illite, chlorite, and ankerite, can be a sink of oxygen during the weathering of sedimentary rocks (Brantley et al., 2013; Sullivan et al., 2016). However, in settings where pyrite minerals are present, the chemical weathering of these other ferrous iron bearing minerals progresses relatively slowly and advances only more rapidly following the complete oxidation of pyrite minerals (Gu et al., 2020a, 2020b). Accordingly, pyrite minerals should be the dominating inorganic $O_2$ sink at the two study sites.

It has been previously suggested that the oxidation of $OC_{petro}$ progresses in a stepwise manner, with the formation of oxygenated compounds of organic matter prior to the release of $CO_2$ (Chang & Berner, 1999), typically resulting in an increase of the relative oxygen content of $OC_{petro}$ during chemical weathering (Longbottom & Hockaday, 2019; Petsch, 2014; Tamamura et al., 2015). If the oxidation of $OC_{petro}$ progresses more rapidly than the separate process of $CO_2$ release from oxygenated $OC_{petro}$ in the Draix-Bléone observatory, this could partly explain the lower ratio of $CO_2$ released compared
to the $O_2$ uptake. However, for rocks exposed in rapidly eroding terrains in the Draix-Bléone observatory, previous studies did not find a significant effect of weathering on the chemical composition of $OC_{petro}$, despite a decrease in the quantities of $OC_{petro}$ and pyrite minerals (Copard et al., 2006; Graz et al., 2011). Thus, it seems unlikely that the oxidation of $OC_{petro}$ at both study sites deviates notably from the theoretical stoichiometry mentioned above.

Furthermore, if the sulfuric acid derived from the oxidation of pyrite minerals interacts with silicate minerals
(Blattmann et al., 2019; Bufe et al., 2021), this would lead to $O_2$ consumption but no $CO_2$ release. In addition, sulfuric acid





could interact with $OC_{petro}$ and be neutralized, yet the vast majority of $OC_{petro}$ is typically made of kerogen, which is resistant to acid hydrolysis, and only minor amounts of more labile organic matter can be prone to this type of degradation (Killops & Killops, 2005; Petsch, 2014; Seifert et al., 2011; Włodarczyk et al., 2018).

685        Another explanation may involve the lateral transport of $CO_2$ as part of the dissolved load, lowering the gaseous release of carbon. The oxidation of $OC_{petro}$ and pyrite minerals coupled to the dissolution of carbonates occur in a humid weathering zone (Fig. 3), where $CO_2$ may be incorporated into the dissolved inorganic carbon (DIC) pool (Bao et al., 2017; Roylands et al., 2022; Soulet et al., 2021; Torres et al., 2014). A recent study has quantified the export of DIC using the molar ratio of $O_2$ and $CO_2$ fluxes for sedimentary rocks undergoing weathering below a forested hillslope (Tune et al., 2020, 2023). There, carbon is sourced from soils, roots and $OC_{petro}$, with an absence of an inorganic carbon source and of pyrite

minerals. If a part of the $CO_2$ from oxidative weathering is exported as DIC in the Brusquet and the Moulin catchments, this would raise the observed $O_2$ consumption to $CO_2$ release ratio, and would have to do so by a factor in the range of ~ 4 and ~ 15. This is worthy of future research towards understanding $O_2$ consumption in the weathering zone. Here, a more accurate quantification is hindered because the proportions of inorganic carbon release versus organic carbon release (which derive from weathering reactions with a different stoichiometry as described above) are unknown. This again calls for future work

to assess the carbon sources in more detail.

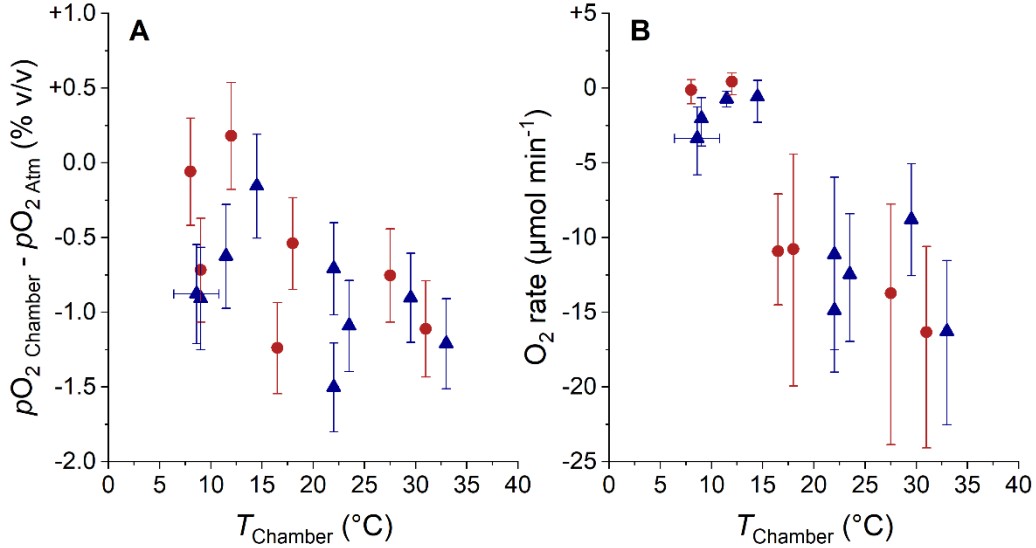

**Figure 12: Comparison of $pO_{2\ Chamber}$ (Panel A, normalized to atmospheric $pO_2$, error bars: RMSE) and of $O_2$ consumption rate (Panel B, error bars: 95 % CI) with chamber temperature. The origin of samples is indicated with red circles for the Brusquet catchment and blue triangles for the Moulin catchment. Estimated temperatures are indicated by accompanying error bars**
**(RMSE).**

       Overall, we have developed the tools needed to quantify the production or consumption, storage and movement of $CO_2$ and $O_2$ in the near-surface of rocks undergoing weathering (Appendix B). If combined, for example, with surface chambers (for gaseous exchange) and boreholes extending below the oxidation front (profiling gaseous and dissolved





processes) (Tokunaga et al., 2016; Tune et al., 2020), alongside radiocarbon and stable carbon isotope analyses (for
partitioning the weathering reactions) (Keller & Bacon, 1998; Roylands et al., 2022; Soulet et al., 2021; Tune et al., 2023),
the cycling of carbon and oxygen in the total critical zone and its environmental controls could be investigated
comprehensively. Using drilled chambers benefits investigations using the gradient method for profiles of gas
concentrations, because with the approach suggested here the diffusivity measures can be determined in situ and in real-time,
which are otherwise typically estimated (Keller & Bacon, 1998; Maier & Schack-Kirchner, 2014; Tokunaga et al., 2016;
Tune et al., 2020). Furthermore, by assessing the release and movement of nitrous oxide from the subsurface (Wan et al.,
2021), an overall greenhouse gas budget could be developed for sedimentary rocks undergoing weathering. This would be
especially valuable for sites with a thin soil cover, which typically dominate more widespread terrains at lower slopes
(Heimsath et al., 2012; Milodowski et al., 2015). There, the additional, modern carbon pool complicates the disentangling of
biogeochemical processes and the corresponding source-specific $CO_2$ and $O_2$ fluxes (Copard et al., 2006; Hemingway et al.,
2018; Keller & Bacon, 1998; Longbottom & Hockaday, 2019; Tune et al., 2020, 2023).

## 5 Conclusions

This study has further developed and assessed methods for in situ constraints on the release of $CO_2$ and the consumption of
$O_2$ during oxidative weathering of exposed sedimentary rocks. Our new method framework allows for both accurate
quantification of weathering fluxes over hourly to daily timescales, while also constraining diffusive processes in the shallow
weathering zone. At two sites of the Draix-Bléone observatory (France), accumulation chambers were installed by drilling
holes directly into rocks undergoing weathering in the Brusquet catchment (black shales) and in the Moulin catchment
(marls). At each site, using an array of 4 chambers, measurements of $pCO_{2\,Rock}$ and $CO_2$ fluxes were carried out alongside
$pO_{2\,Rock}$ measurements during six fieldtrips over one year.

We find that during a single visit to a chamber, the accumulation rates decline over a few measurement cycles,
before reaching a stable $CO_2$ accumulation rate. This pattern is consistent across the fieldtrips and can be described by an
exponential model. To explain these observations, we outline a framework which considers the measured $CO_2$ accumulation
as a combination of the real-time production during weathering, plus the release of excess $CO_2$ built up in pore space
surrounding the chambers. By doing so, we can assess the rock pore volume and rock mass that produce $CO_2$. For the first
time, this allows an absolute report of rock-derived $CO_2$ fluxes measured in situ and in real-time, providing input data for
future studies modeling the chemical weathering of sedimentary rocks. The assessment of contributing rock pore space
allows us also to normalize the fluxes to an outcrop surface area, enabling comparison of the weathering fluxes at the study
sites to other rock types and soils across different terrains and climates. Furthermore, by studying the accumulation of $CO_2$ in
a chamber and the connected rock pore space over time, the diffusivity of gases in the shallow weathering zone and its
environmental controls are investigated, including an absolute, in situ determination of the diffusion coefficients.



In addition to these insights into the $CO_2$ release, $pO_2$ values for the studied rocks are presented and used together with the quantification of the diffusive processes in the weathering zone to calculate $O_2$ fluxes. It is shown that the consumption of $O_2$ co-varies with changes in the emission of $CO_2$ over time, which are driven by changes in temperature. However, the $O_2$ fluxes indicate significantly greater oxidative weathering rates compared to the $CO_2$ fluxes. We suggest this discrepancy results of the export of inorganic carbon by the dissolved load of percolating waters lowering the effective

release of gaseous $CO_2$.

        A site-specific difference in the magnitude of $CO_2$ emissions at the two study sites cannot be explained by differences in the lithological properties influencing the diffusion of gas within the rock space surrounding the chambers as both study sites have similar characteristics, which is evidenced by diffusivity measures changing similarly alongside temperature and precipitation. This finding suggests that differences in the source of carbon are the main reason for the

observed $CO_2$ flux differences, providing an opportunity for future research to investigate the control of the chemical composition of the rocks (i.e., contents of $OC_{petro}$, carbonates and sulfides) on the $CO_2$ flux size.

**Appendices**

**Appendix A - Modeling chamber temperatures**

To fill the gaps in the direct chamber temperature measurements, we use air temperatures from a local weather station as a

proxy by modifying a framework that describes soil temperatures by, amongst other variables, air temperature (Liang et al., 2014). The approach combines measured air temperatures with a Fourier-fitted function that describes the daily average temperature inside the rock chambers by weighting averaged air temperatures by the fractional duration of daylight ($L$) at the latitude of the Draix-Bléone observatory. In more detail, we estimate the current chamber temperature $T_{Chamber}$ (°C) at an hourly resolution as follows:

$$T_{Chamber} = T_{mean} \times coeff_A + ( T_{Air-6h} - T_{mean}) \times coeff_B ,  \tag{A1}$$
        where $T_{Air-6h}$ is the hourly air temperature (°C) from nearby meteorological stations (Draix-Bléone Observatory, 2015) delayed by 6 hours, $coeff_A$ and $coeff_B$ are site-specific fitting coefficients, and $T_{mean}$ (°C) is the long-term trend of rock temperature described by:

        $$T_{mean} = coeff_{C1} + coeff_{C2} \times cos(coeff_{C3} \times T_{air,7d} \times L) + coeff_{C4} \times sin( coeff_{C3} \times T_{Air, 7d} \times L) ,  \tag{A2}$$

where $coeff_{C1}$ to $coeff_{C4}$ are site-specific fitting coefficients (Table A1) in a 1st order Fourier-model, and $T_{Air,7d}$ (°C) is the 7-day average of the past air temperatures at hourly resolution. Using the site-specific air temperatures, this approach simulates $T_{Chamber}$ well, with a root-mean-square error (RMSE) of 1.8 °C for the Brusquet catchment and 2.2 °C for the Moulin catchment (Fig. A1).



Figure A1: Panels A: comparison of chamber temperatures measured and estimated by a modeling framework based on air temperature and the fractional duration of daylight at the latitude of the Draix-Bléone observatory at hourly resolution for the Brusquet catchment (red) and the Moulin catchment (blue), which agree with a 1:1 relation (dashed line). Panels B: normally distributed residuals between the measured temperatures and the modeling framework.

Table A1: Overview of site-specific fitting coefficients used for modeling chamber temperatures based on air temperature and fractional duration of daylight (Eq. A1 and A2) and details of the goodness of the fitting model based on comparisons to measured chamber temperatures with hourly and daily resolution.

| Site | $R^2$ | p-value | n | RMSE (hourly) (°C) | RMSE (daily) (°C) | $coeff_A$ | $coeff_B$ | $coeff_{C1}$ | $coeff_{C2}$ | $coeff_{C3}$ | $coeff_{C4}$ |
|---|---|---|---|---|---|---|---|---|---|---|---|
| Brusquet | 0.96 | <0.001 | 7,392 | 1.80 | 1.58 | 1.147 | 0.361 | 0.408 | 2.364 | 0.080 | 30.106 |
| Moulin | 0.94 | <0.001 | 5,664 | 2.21 | 1.99 | 1.222 | 0.298 | 0.362 | 2.074 | 0.093 | 26.407 |



## Appendix B - Data-flow diagram for chamber-based CO₂ and O₂ flux measurements



**Figure B1: Central data-flow for the new approach developed in this study to quantify the diffusive exchange of CO₂ and O₂ during shallow rock weathering based on real-time measurements using drilled headspace chambers.**





**Appendix C - Linear regression of $p\text{CO}_{2\,\text{Rock}}$ and $\text{CO}_2$ accumulation rates**

**Table C1: Details of linear regressions comparing measurement-specific values of $p\text{CO}_{2\,\text{Rock}}$ and $\text{CO}_2$ accumulation (based on repeats 6 - 8) (with $p\text{CO}_2\,(ppmv) = a_1 \times \text{CO}_2$ rate ($\mu$gC min$^{-1}$) + $a_2$) including the standard errors of the fitting parameters. Differentiations into "dry" and "wet" samples are based on a threshold of a cumulative precipitation of 5 mm over 3 days prior to**
**the measurement. Both sites have similar linear regressions with overlapping standard errors.**

| Data set | $a_1$ | $a_2$ | $R^2$ | p-value | n |
|---|---|---|---|---|---|
| all | $55.1 \pm 5.1$ | $848.7 \pm 83.5$ | 0.69 | <0.001 | 55 |
| *Brusquet* | *$55.0 \pm 6.3$* | *$741.6 \pm 120.5$* | *0.77* | *<0.001* | *25* |
| *Moulin* | *$61.1 \pm 8.5$* | *$872.5 \pm 117.0$* | *0.64* | *<0.001* | *30* |
| all - dry | $59.1 \pm 6.4$ | $664.3 \pm 122.2$ | 0.75 | <0.001 | 31 |
| all - wet | $61.0 \pm 8.3$ | $948.7 \pm 101.0$ | 0.71 | <0.001 | 24 |

**Appendix D - Linear regression of diffusivity measures**

**Table D1: Details of linear regressions comparing the diffusivity measures $\frac{D(\text{CO2})}{\omega}$ and $\lambda$ (with $\frac{D(\text{CO2})}{\omega}$ (cm$^3$ min$^{-1}$) = $b_1 \times \lambda$ (min$^{-1}$) + $b_2$) including the standard errors of the fitting parameters. Differentiations into "dry" and "wet" samples are based on a threshold of a cumulative precipitation of 5 mm over 3 days prior to the measurement. The regressions are based on using the**
**average values from the repeats 6 - 8 of the flux measurements, which is typically close to stabilization of the $\text{CO}_2$ accumulation rate, because a significant number of the measurements was limited to 8 repeats. Both sites have comparable linear regressions with overlapping standard errors.**

| Data set | $b_1$ | $b_2$ | $R^2$ | p-value | n |
|---|---|---|---|---|---|
| all | $163.4 \pm 13.2$ | $-1.13 \pm 2.25$ | 0.74 | <0.001 | 55 |
| *Brusquet* | *$162.1 \pm 13.9$* | *$-1.54 \pm 2.67$* | *0.86* | *<0.001* | *25* |
| *Moulin* | *$170.4 \pm 24.0$* | *$-1.57 \pm 3.61$* | *0.64* | *<0.001* | *30* |
| all - dry | $151.4 \pm 18.5$ | $2.25 \pm 3.58$ | 0.71 | <0.001 | 31 |
| all - wet | $157.2 \pm 22.9$ | $-1.94 \pm 3.13$ | 0.68 | <0.001 | 24 |

**Appendix E - Different diffusion pathways of a closed versus manipulated chamber**

The diffusion pathways of a closed chamber differ from that of a manipulated chamber. During a flux measurement, $\text{CO}_2$, which is released from a rock grain undergoing oxidative weathering into the pore space, moves via diffusion along a concentration gradient, which is initiated by lowering repeatedly the $p\text{CO}_{2\,\text{Chamber}}$ to a near-atmospheric level (Soulet et al., 2018), from the rock pores towards the manipulated chamber if they are connected more effectively to the chamber than to the rock-atmosphere boundary. In contrast, without a sampling system acting as the receiving reservoir, rock-derived $\text{CO}_2$
travels along the concentration gradient towards the atmosphere. In the latter scenario, rock pores that contribute $\text{CO}_2$ during a flux measurement to a chamber either contribute the $\text{CO}_2$ via diffusion directly to the atmosphere (without a pathway through the chamber) or through the chamber towards the atmosphere.

**Appendix F - Repeated CO$_2$ flux measurements on the same date**

On four visits, a chamber was measured twice a day and the observed CO$_2$ release was higher in the afternoon than in the morning, coinciding with an increase of the chamber temperature: for chamber 5 at the Brusquet site, $q_{Plateau}$ increased from 1.1 µgC min$^{-1}$ to 3.8 µgC min$^{-1}$ and from 2.6 µgC min$^{-1}$ to 4.7 µgC min$^{-1}$ coinciding with temperate increases from -0.9 °C to 1.0 °C and from 2.0 °C to 5.0 °C, respectively; for chamber 4 at the Moulin site, $q_{Plateau}$ increased from 14.6 µgC min$^{-1}$ to 18.0 µgC min$^{-1}$ and from 16.3 µgC min$^{-1}$ to 21.4 µgC min$^{-1}$ coinciding with temperate increases from 22.5 °C to 26.5 °C and from 22.0 °C to 27.0 °C, respectively.

**Data availability**

All data supporting the findings of this study will be uploaded to a data repository. For the review process, these data can be found in supplementary tables.

**Author contribution**

The research was conceptualized by TR, with help from RGH and ELM. The main methods were designed by TR, with help
from RGH, GS, MHG and ELM. Field-based and laboratory geochemical measurements were led by TR, with contributions from MHG, RGH, SK, MD and FN, and climate data and aerial imagery were provided by SK and CLB. TR led the formal analysis, investigation, data visualization and writing of the original draft, under the supervision of RGH and ELM. All authors contributed to subsequent review and editing, led by TR, RGH, ELM and GS. Funding was acquired by RGH, TR and MHG.

**Competing interests**

The authors declare that they have no conflict of interest.

**Acknowledgements**

This work was supported by the European Research Council (Starting Grant to RGH, ROC-CO2 project, grant 678779). Analytical work was also supported in part by the Natural Environment Research Council (NERC, UK; NEIF Radiocarbon
Grant to TR, RGH and MHG, grant 2201.1019).



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
