# Peer review of "Probing the exchange of $CO_2$ and $O_2$ in the shallow critical zone during weathering of marl and black shale"

_Earth Surface Dynamics, 2023_

## Referee Comment (RC1)

Review of *Probing the exchange of CO2 and O2 in the shallow critical zone during weathering of marl and black shale* submitted to Earth Surface Dynamics

I found this paper to be a very interesting, detailed, and innovative exploration of how to quantify $CO_2$ production from oxidative weathering in situ. To me, the paper is clearly written and well organized. Admittedly, the details of the diffusion calculations were not always very straightforward for me to follow, but that probably lies more in my lack of background than in unclear writing. I only have a few minor comments that I invite the authors to consider before publication.

**Calculation and significance of contributing rock-volumes**

In L515ff, you calculate the contributing volumes. Do these calculations of the contributing rock volume assume that that all of the 1216 cm$^3$ (or the 1.9 cm length) contribute equally? Somehow, I would have expected a decay in the contribution with distance from the chamber where the closest rock contributes most and then the rock farther away contributes less. Then, the length or volume would be some characteristic length-scale (or volume-scale) that describes the decay?

What are the significance of these (short) length-scales of $CO_2$ diffusion for the weathering of an entire outcrop or soil profile? Would you expect, form your results, that only the topmost few centimeters contribute to direct degassing into the atmosphere and the products from deeper weathering pathways go into the groundwater? This question touches on the previous one, wondering about the length scales over which CO2 production is relevant

**Line comments**

L40: The citations here (Gaillardet et al., 1999; Moon et al., 2014) only justify the global CO2 drawdown flux from silicate weathering, they do not support the first part of the sentence which also warrants a citation, I think.

L123/Figure 1: To make the figure even clearer, I suggest to either add the catchment names on the map, or the colored circles on the sub-catchments in the legend as well

L131: "To measure in situ the production of CO2 […]". Here, you could specify where you (want to) measure the fluxes (in bedrock, regolith, soil etc.).

L135: Which shape?

L136: Perhaps "Install the chambers *at*"?

L138: Here you switch from past tense in the previous sentences to present tense in this and the following section. Suggest to keep consistent

L181: "with respect to"?

L192: "the curvature of the mass change" is unclear to me. Is the curvature of the mass change the second derivative of the mass change with respect to time? That would mean the third derivative of mass with respect to time?

L193: When you say "For this" what does "this" refer to? Do you use equation (3) to calculate the variable m(t) in equation (2)? If yes, is there one of the variables in equation (3) that is a function of time (Probably $pCO_2$)? That could be marked with *(t)* to clarify.

L238: Could you describe a bit more how you get from Fick's law (equation 1) to this expression of the diffusive flux? In particular, I do not quite understand the variable ω. First, you write that it describes a diffusion "over depth and area", but the units ($cm^1$ $cm^{-2}$) look like "a depth over an area". Can the meaning of this spatial scale be explained in a sentence? Also, this variable has units of an inverse length $[L^{-1}]$ whereas dz in equation (1) has units of a length $[L]$ – so, it seems that the units of the resulting flux are different (it is possible that I am making an algebraic mistake somewhere)? Could you comment on the link to equation (1)?

L283: Could you specify how to get SR>1? This would imply that you recover more $CO_2$ in the lab than you measure in the field. I guess that is only possible because of uncertainties in the measurements, or could you have gained some $CO_2$ elsewhere?

L288: could you specify "change in x-axis scale" (which is what you refer to, I think?).

L290: switch to past tense? ("consisted")?

L298: You could refer to a figure or table to substantiate the statement.

L368 - 370: As far as I understand, these two sentences note two contrasting observations ($CO_2$ production maximized either during dry or during wet conditions). In L371, the sentence starts with "This observation". Can you clarify which of the two options you refer to?

Also, more importantly, I didn't walk away from the paragraph with an understanding what could cause these contrasting observations or whether that is even still an open question. The following explores how diffusivities should be lower during wet conditions but that can only explain one of the observations – at least I think?

L417 & 421: In both cases, that should be Fig. 6I, I think?

L461: Should there be a "dt" in the equation?

L498: Maybe "The coincidence also *suggests* […]"?

Fig9: I wonder if the trends in the data would be clearer with a logarithmic x-axis?

L556: Is it possible to convert this number into an estimate of equivalent CO2 production in terms of tons per square kilometer per year, a number that readers may be familiar with?

L593: I think I know what the dark and light colors mean, but I can imagine that some readers might miss the link to what you mean by "extrapolated" or "stabilized". You could refer here to figure 5 or L339f.

L736-737: Did I miss something, or did you give a reason for why you did not directly contrast your $O_2$ consumption rates (Figure 12) with the corresponding $CO_2$ accumulation rates?

L738ff: Why do you not mention silicate weathering with sulfuric acid as the alternative option that you had discussed earlier? If you gave a reason for dismissing it, I missed it.

I hope that the comments are helpful, and I remain with best wishes to the authors and editor.

Aaron Bufe

**References**

Gaillardet, J., Dupré, B., Louvat, P., and Allègre, C. J., 1999, Global silicate weathering and CO2 consumption rates deduced from the chemistry of large rivers: Chemical Geology, v. 159, no. 1, p. 3-30.

Moon, S., Chamberlain, C. P., and Hilley, G. E., 2014, New estimates of silicate weathering rates and their uncertainties in global rivers: Geochimica et Cosmochimica Acta, v. 134, p. 257-274.

---

## Author Comment (AC1)

**Author comment (AC) on esurf-2023-15 (manuscript titled "*Probing the exchange of CO₂ and O₂ in the shallow critical zone during weathering of marl and black shale*")**

by Tobias Roylands on behalf of all authors

**1 Preface**

Dear editor, dear referees, dear community,

please find below our reply to the referee comments (RC) to our manuscript submitted to Earth Surface Dynamics (esurf-2023-15). We are very thankful for gathering the reviews and for the positive and helpful feedback from the referees. In line with their comments, we have made some changes to the manuscript throughout. A detailed response to all referee comments is given below, with the referee comments provided in *black and italic* and with our responses in blue. All changes to the manuscript are documented here (first section with page numbers starting with i) and visualized in a further section below (with page numbers starting with 1) using the "track changes function".

Many thanks again.
Sincerely,

Tobias Roylands on behalf of all authors

**1 Response to RC1 by Aaron Bufe (29 Jun 2023)**

*I found this paper to be a very interesting, detailed, and innovative exploration of how to quantify CO₂ production from oxidative weathering in situ. To me, the paper is clearly written and well organized. Admittedly, the details of the diffusion calculations were not always very straightforward for me to follow, but that probably lies more in my lack of background than in unclear writing. I only have a few minor comments that I invite the authors to consider before publication.*
Many thanks for your comments and questions which have helped us improve our work.

***Calculation and significance of contributing rock-volumes***
*In L515ff, you calculate the contributing volumes. Do these calculations of the contributing rock volume assume that that all of the 1216 cm3 (or the 1.9 cm length) contribute equally? Somehow, I would have expected a decay in the contribution with*

*distance from the chamber where the closest rock contributes most and then the rock farther away contributes less. Then, the length or volume would be some characteristic length-scale (or volume-scale) that describes the decay?*

The calculation of the contributing rock volume indeed assumes that all rock pores (and the locally associated minerals) of the calculated volume contribute to the chamber-based fluxes, but the approach cannot assess their relative contributions. In other words, we cannot assess how the contribution scales with distance from the chamber. For example, after starting a flux measurement, closer pores (i.e., pores with a shorter effective diffusion pathway to the chamber) will contribute their "excess" $CO_2$ earlier than more distant pores. As excess $CO_2$ is removed during the flux measurement process, more distant pores may become more important. In the chamber, the real-time accumulation of $CO_2$ will only reach a plateau ($q_{Plateau}$) once this excess $CO_2$ has been exhausted from across the connected pore volume. For this reason, the assumption that all rock pores connected to a chamber contribute is valid because of the hour-long timescales and evolution of flux measurements that we observe, and we do not need to know their relative contribution. We have clarified Section 4.1.1 accordingly to note this.

To take the research further, it would be very interesting to try and quantify the micro-structure of the studied rocks in detail. Based on field measurements of the physical properties of the rocks, we can only assume on the geometry of the contributing rock volume (Sections 4.2.2 and 4.2.3). The diffusion pathways of different pores within the contributing rock volume will be set by their distance to the chamber, but also perhaps more importantly by the micro-structure of the rock. Indeed, this may include fractures, joints and bedding surfaces which act as pathways for gases. These factors are why we focus on a parameter which combines depth and area for most of the text (i.e., $\omega$). In contrast, for an in situ estimate of the overall diffusivity in the contributing rock volume in Section 4.2.3, the chamber inner surface area is used in combination with the probed layer thickness to calculate $\omega$ to obtain $D$ from $\frac{D(CO2)}{\omega}$.

*What are the significance of these (short) length-scales of $CO_2$ diffusion for the weathering of an entire outcrop or soil profile? Would you expect, from your results, that only the topmost few centimeters contribute to direct degassing into the atmosphere and the products from deeper weathering pathways go into the groundwater? This question touches on the previous one, wondering about the length scales over which $CO_2$ production is relevant.*

This is an interesting question, but not one we can expand on any further based on our available data. When we assess the contributing rock volume and its $pCO_2$, we find that it can derive from a ~ 2.2 cm thickness around the studied chambers. However, this is a minimum estimate, because without more information about the physical structure of the pore space (see our reply above), the oxidative weathering zone is treated as one homogeneous layer.

As to whether deeper $CO_2$ could be exported by subsurface water flow, more detailed in situ measurements of the solid, dissolved and gaseous evolution of the shallow critical zone will be needed, alongside those that track precipitation migrating through the shallow critical zone, to address this interesting and relevant research gap. That said, our $CO_2$ and $O_2$

mass balance does suggest some export of C out of the weathering zone in dissolved form, and so our manuscript already discusses these themes and we have not expanded on them.

*Line comments*

*L40: The citations here (Gaillardet et al., 1999; Moon et al., 2014) only justify the global $CO_2$ drawdown flux from silicate weathering, they do not support the first part of the sentence which also warrants a citation, I think.*
Agreed. According references have been added to the text.

*L123/Figure 1: To make the figure even clearer, I suggest to either add the catchment names on the map, or the colored circles on the sub-catchments in the legend as well*
Agreed. Colored circles have been added in the legend as well.

*L131: "To measure in situ the production of $CO_2$ [...]". Here, you could specify where you (want to) measure the fluxes (in bedrock, regolith, soil etc.).*
Agreed. More detail has been added to the text.

*L135: Which shape?*
The text now refers to the shape of the drilled chambers.

*L136: Perhaps "Install the chambers at"?*
The text now reads "To install the chambers to a depth of ~ 38 cm,…" to clarify that they "start" at the surface. A reference to Fig. 2E has been added as well.

*L138: Here you switch from past tense in the previous sentences to present tense in this and the following section. Suggest to keep consistent*
Agreed.

*L181: "with respect to"?*
Agreed.

*L192: "the curvature of the mass change" is unclear to me. Is the curvature of the mass change the second derivative of the mass change with respect to time? That would mean the third derivative of mass with respect to time?*

The curvature reflects a constant covering the sum of all processes that are proportional to the carbon mass difference ($m(t)$ - $m_0$) and that relate to the diffusion of $CO_2$ between rock pores, chamber and atmosphere. The text has been clarified accordingly.

*L193: When you say "For this" what does "this" refer to? Do you use equation (3) to calculate the variable m(t) in equation (2)? If yes, is there one of the variables in equation (3) that is a function of time (Probably pCO₂)? That could be marked with (t) to clarify.*

Yes, this is the case. The equation has been modified for clarification.

*L238: Could you describe a bit more how you get from Fick's law (equation 1) to this expression of the diffusive flux? In particular, I do not quite understand the variable ω. First, you write that it describes a diffusion "over depth and area", but the units ($cm^1$ $cm^{-2}$) look like "a depth over an area". Can the meaning of this spatial scale be explained in a sentence?*

*Also, this variable has units of an inverse length $[L^{-1}]$ whereas dz in equation (1) has units of a length $[L]$ – so, it seems that the units of the resulting flux are different (it is possible that I am making an algebraic mistake somewhere)? Could you comment on the link to equation (1)?*

Please note the response to the first comment above. Furthermore, we note after looking into this in more detail that we found a small but important typo in Equation 1 with the variable $j$ was given earlier instead of $J$ which resulted in a missing surface (compare Equation 5) and explains the mentioned discrepancy of parameters and units.

In more detail, $\omega$ combines i) the surface of the flux with ii) the depth over which the concentration change is studied. Both relate to the unknown, heterogeneous and complex structure of the contributing rock pore space beginning at the chamber walls. We have modified Equation 1 and the description of Equation 7 accordingly to help clarify this point.

*L283: Could you specify how to get SR>1? This would imply that you recover more CO₂ in the lab than you measure in the field. I guess that is only possible because of uncertainties in the measurements, or could you have gained some CO₂ elsewhere?*

Indeed, a SR > 1 can be explained by uncertainties in the measurements. Recovery of $CO_2$ standards from the traps used in this study is robust, with a 95% recovery (see cited work in the study and Sect. 4.1.1). The uncertainty therefore likely relates most to the determination of $V_{CO2-MSC}$ (i.e., the determination of the difference in $pCO_{2\ Chamber}$ during sampling; (e) in Fig. 4). To make this point easier to ascertain, we now give the average alongside the standard deviation.

*L288: could you specify "change in x-axis scale" (which is what you refer to, I think?).*

Agreed.

*L290: switch to past tense? ("consisted")?*

The caption now reads in present tense.

*L298: You could refer to a figure or table to substantiate the statement.*

Agreed. We refer now to Fig. 12A.

*L368 - 370: As far as I understand, these two sentences note two contrasting observations ($CO_2$ production maximized either during dry or during wet conditions). In L371, the sentence starts with "This observation". Can you clarify which of the two options you refer to? Also, more importantly, I didn't walk away from the paragraph with an understanding what could cause these contrasting observations or whether that is even still an open question. The following explores how diffusivities should be lower during wet conditions but that can only explain one of the observations – at least I think?*

The observation mentioned in the 2nd sentence was referred to. However, please note that the observations are not conflictive but complementary. The $pCO_{2\ Rock}$ allows consideration of the storage of $CO_2$, which provides - together with the $CO_2$ production (i.e., flux) - a means to investigate the diffusion as discussed in the following lines. To better explain this complementary nature, the overall paragraph has been clarified.

*L417 & 421: In both cases, that should be Fig. 6I, I think?*

Agreed. Similar mistakes have been corrected in Section 3.1.

*L461: Should there be a "dt" in the equation?*

Agreed. The equation has been modified accordingly.

*L498: Maybe "The coincidence also suggests [...]"?*

Agreed.

*Fig9: I wonder if the trends in the data would be clearer with a logarithmic x-axis?*

Indeed. Fig. 9 and its caption have been updated accordingly.

*L556: Is it possible to convert this number into an estimate of equivalent $CO_2$ production in terms of tons per square kilometer per year, a number that readers may be familiar with?*

Please note the fluxes (using this unit) that are given two paragraphs above. There, we attempt a conversion of the contributing rock volume based on an assumption on the thickness of the weathering zone that contributes to the fluxes, but we note the uncertainty on this thickness. In contrast, the rock mass (also based on the contributing rock volume) is provided later and independently from the weathering depth (please also note the central data-flow in Fig. B1).

*L593: I think I know what the dark and light colors mean, but I can imagine that some readers might miss the link to what you mean by "extrapolated" or "stabilized". You could refer here to figure 5 or L339f.*

Agreed.

*L736-737: Did I miss something, or did you give a reason for why you did not directly contrast your $O_2$ consumption rates (Figure 12) with the corresponding $CO_2$ accumulation rates?*

Please note the discussion in Section 4.4 regarding the direct comparison of $O_2$ and $CO_2$. In addition, we have added a panel with a comparison to Fig. 12 (i.e., new Fig. 12C).

*L738ff: Why do you not mention silicate weathering with sulfuric acid as the alternative option that you had discussed earlier? If you gave a reason for dismissing it, I missed it.*

Thanks. This has been corrected and the conclusion now refers to silicate weathering with sulfuric acid as well.

*I hope that the comments are helpful, and I remain with best wishes to the authors and editor.*
*Aaron Bufe*

*Other comments:*

*Figure 4 is well done and very instructive.*
Many thanks.

*You may want to check the paper for issues with significant figures (e.g. L556).*
We've fixed this issue, and have checked elsewhere.

*Are the ranges of values for tC/km2/year that appear in the abstract associated with the values in line 458? If so, please add the range of values to the text and help readers connect Table 5 to this range.*
Thanks for spotting this. The flux-values in the abstract are associated with the values in the former line 548. Now, the corresponding ranges were added to the main text as well. These are given based on averages for minima and maxima of the accumulation rates in Table 5 and they now consider also the chamber-specific values for the contributing rock volume – according clarification has been added to the text and the values have been updated because of this and comments below (with respect to the weathering depth and to the porosity).

*I commend the authors for leveraging well studied sites. I found it a bit unconventional to cite a list of publications for relatively straight forward site attributes (e.g. L117, L120). This makes it harder to identify the source of the information cited. I also wonder whether there are stream/surface water fluxes of DIC or major cations that could be used to comment on the feasibility of the dissolution and transport of CO₂ (i.e. what is proposed in L8).*
We agree that the details of the study site are not always straight forward to extract from the references cited in Sect. 2.1. As such, the review by Mathys & Klotz (2008) has been placed earlier in the section, giving a first reference for more detailed reading. Otherwise, this section summarizes all the information that is relevant for our study while citing all sources and trying to separate topic-source-combinations as much as possible. In most cases, without extending the text length or going into more detail, it is not possible to further separate the "list of publications". This is also true for the former L117. The former L120, however, has been modified to add more clarity as suggested.

*L51: This could be written more clearly to support the motivation for the work presented here. Tune et al. use the gradient method as well as water chemistry to demonstrate that deep CO₂ is associated with deep roots in bedrock. They find that the solid phase is depleted in OC$_{petro}$ but that the CO₂ that is transiting upwards from the rock is not derived from oxidation of*

$OC_{petro}$ *(via $^{14}C$.) Note also that Tune et al., 2023 perform rock incubation experiments and report lab-based oxidation rates. These could be compared to the fluxes reported here.*

To clarify the motivation, more detail of the approach of Tune et al. has been added in the introduction. Their findings will be most relevant for studies partitioning total $CO_2$ fluxes to study the fate of $OC_{petro}$ during oxidative weathering. However, without partitioned fluxes (beyond the scope of this study – please note replies to RC2 above) a comparison to the field- and lab-based oxidation rates by Tune et al. is not reliably feasible right now. We look forward to future work which can make this link between the growing number of field measurements and lab incubations.

*L85: Is this in reference to solid phase measurements of $OC_{Petro}$? Could clarify.*

Yes. This has been clarified accordingly.

*L112- Could you clarify what is meant by "more comparable between the catchments?"*

This was meant with respect to similarly high erosion rates. The text has been clarified accordingly.

*L365-375: This section was a bit challenging to parse and could be clarified to separate hypotheses from observations. As is written, it is well established that diffusive fluxes would decrease under reduced air filled porosity and increased tortuosity associated with moisture increase. It would be helpful to spell out other contributions to changes in observed fluxes more clearly. For example, later in the paper we learn about solifluction and solid phase changes which could be described here instead.*

To better separate hypotheses from observations, this section has been clarified – also in line with RC1 (please see above). However, content has not been moved within the overall Section 4.1.2. to keep the step-by-step increase of detail and corresponding explanation.

*L384- remove likely? Under what circumstances might this not be the case? Also, consider revisiting Sanchez Canete et al., 2018 and citing here.*

As suggested, "likely" has been removed. The suggested reference is helpful, thank you, and has been added here as well as in the beginning of Section 4.4.

*While I don't propose a major overhaul on the structure of the paper, I did find it a bit unconventional and a bit of a challenge to work through all of the calculations in the discussion. The paper would perhaps be more accessible and easier to read if the methods and calculations were all spelled out in the methods section for easier reference. This is ultimately at the author's discretion.*

We understand that the structure of this manuscript may seem somewhat unconventional. Because we lack oxidative weathering field measurements, a purely methods paper would not be appropriate because we have learnt so much more

about the shallow weathering zone in our study. Indeed, in an early version of the manuscript, the structure suggested by the reviewer was tried. It was dropped by the authorship team because various steps of the new method provide "intermediate" results that require an explanation before adding more detail and complexity with the later steps of the method. As such, we have decided not to make this change in structure and hope that Fig. B1 also provides an overview for the reader in this mixture of method development, new field measurements and discussion paper.

*Figure 6 and associated text: Were there indications of which events led to infiltration to corroborate the 5 mm for 3 day threshold? Did surveyors observe other evidence of subsurface wetting that aided in the classification of data between wet and dry?*

This threshold is used as an approximation to distinguish "wet" from "dry" settings allowing investigation of the effect of precipitation. This approximate threshold is based on the precipitation record and on field-based observations when significantly lower precipitation values dried rapidly off the surface. The captions of Fig. 6 and Fig. 9 have been clarified with respect to the approximation.

*L512: 30% air filled porosity seems very high for a fractured rock. I was not able to find the porosity and saturation measurements that this was based on in the papers cited. Can you present the evidence to corroborate these very high air-filled porosity values, or if not, could you justify them and provide clarification on how they affect the results?*

Indeed, porosity values are very high in the weathering zones of the study area while the saturation varies significantly over time. The study by Mallet et al. (2020) provides both, water content and porosity measurements (compare, for example, their Table 1 and Fig. 6), while Garel et al. (2012) provide porosity values (in their study site description) for material from the ablation zone studied by them. However, following a revisit of these references, we now assume an average air-filled porosity of 25 ± 10 %. The according sentence has been both clarified and updated. Furthermore, values in Section 4.2.3 (with respect to the contributing rock volume and rock mass, diffusivity and surface efflux) needed to be recalculated based on this correction, which also results in updated fluxes in the abstract.

*L530: Are these rocks fractured?*

The cited reference does not provide this detail. We have added a corresponding comment in brackets to the sentence.

*L536-541: I was not able to follow how the cited papers support the $Z_{rock}$ calculation. This section seems a bit contradictory with the site description- could you clarify the text to address this?*

In the site description (Sect. 2.1), a more general view with respect to the physical alteration and structure of the critical zone is provided, whereas here an estimate is given for the thickness of the oxidative weathering zone, which is assumed to roughly equal the depth of significant physical alteration. For this, please note that $z_{Rock}$ is not calculated but assumed to equal the mentioned alteration depth.

However, we agree this paragraph could have been much clearer. Indeed, by revisiting this, we have concluded that the previously considered range of $20 \pm 10$ cm does not fully capture the available measurements and our own field observations. As such, we have made some extensive changes in this section including the description and valuation of $z_{Rock}$ (with a link to Sect. 2.1) and the calculation of the $S_{Rock}$ value and topographic surface fluxes (here and in the abstract).

*L555-558: Why aren't rock masses reported separately for the two catchments given that $CO_2$ fluxes are reported separately?*
This has been corrected.

*Sánchez-Cañete, E.P., Barron-Gafford, G.A. and Chorover, J., 2018. A considerable fraction of soil-respired $CO_2$ is not emitted directly to the atmosphere. Scientific Reports, 8(1), p.13518.*

**3 Further minor changes**

- Figure 10 has been updated so that the caps of the error bars fit to the style of the other figures;
- References have been updated with respect to the ESurf-style (not marked in the following section of tracked changes);
- the data availability section now refers to the Supplementary;
- the abstract has been clarified with respect to the oxygen consumption by carbonate dissolution coupled to sulfide oxidation;
- a missing unit was added in the beginning of Section 3.2 for $pCO_{2\,Rock}$ in the Moulin catchment;
- a typo with respect to the unit of $q$ in the caption of Figure 4 has been corrected;
- Figure B1 has been updated for clarification of $V_{CO2\text{-}IRGA}$.

[revised manuscript text omitted]
 | $222 \pm 166$ | 5 | $34^{+44}_{-24}$ | $444^{+78}_{-44}$ | $493 \pm 279$ | 5 | $120^{+153}_{-85}$ | $866^{+424}_{-244}$ |
| 6 | B-G-6 | Brusquet | $353 \pm 267$ | 12 | $64^{+130}_{-37}$ | $863^{+1,709}_{-505}$ | $339 \pm 180$ | 12 | $123^{+223}_{-64}$ | $666^{+286}_{-183}$ |
| 7 | B-H-7 | Brusquet | $142 \pm 126$ | 4 | $20^{+66}_{-16}$ | $336^{+1,352}_{-222}$ | $272 \pm 182$ | 4 | $71^{+229}_{-54}$ | $537^{+2159}_{-354}$ |
| 8 | B-I-8 | Brusquet | $261 \pm 147$ | 7 | $37^{+24}_{-16}$ | $450^{+164}_{-115}$ | $370 \pm 157$ | 7 | $103^{+67}_{-46}$ | $530^{+697}_{-262}$ |
| *Brusquet totals* | | | *277 ± 221* | *28* | | | *365 ± 208* | *28* | | |
| 1 | M-C-1 | Moulin | $252 \pm 150$ | 10 | $10^{+57}_{-10}$ | $453^{+152}_{-113}$ | $280 \pm 122$ | 10 | $38^{+210}_{-38}$ | $408^{+137}_{-102}$ |
| 2 | M-A-2 | Moulin | $55 \pm 26$ | 4 | $15^{+17}_{-11}$ | $79^{+144}_{-42}$ | $164 \pm 72$ | 4 | $60^{+65}_{-42}$ | $247^{+449}_{-133}$ |
| 3 | M-D-3 | Moulin | $217 \pm 50$ | 3 | $149^{+119}_{-62}$ | $267^{+59}_{-46}$ | $256 \pm 48$ | 3 | $189^{+150}_{-79}$ | $293^{+64}_{-51}$ |
| 4 | M-B-4 | Moulin | $296 \pm 192$ | 15 | $48^{+62}_{-26}$ | $569^{+544}_{-232}$ | $404 \pm 193$ | 15 | $145^{+184}_{-77}$ | $744^{+238}_{-169}$ |
| *Moulin totals* | | | *245 ± 174* | *32* | | | *322 ± 174* | *32* | | |

[revised manuscript text omitted]

**Rock-derived $CO_2$ flux measurement using a drilled chamber**

At start:
$$pCO_2{}_{\text{Chamber}} = pCO_2{}_{\text{Rock}}$$
&
$$pO_2{}_{\text{Chamber}} = pO_2{}_{\text{Rock}}$$

During the repeated manipulation (i.e., removal of $CO_2$ followed by accumulation events; Fig. 4):
$q_i$ ($CO_2$ accumulation rate; Eq. 2)
&
$\lambda_i$ (curvature of a single accumulation event; a diffusivity measure; Eq. 2)
&
$V_{\text{CO2-IRGA}}$
(sum of $pCO_2{}_{\text{Chamber}}$ maxima of the acc. events)

$q_1$ to $q_n$

Initial decline but stabilization over time of $q_i$ (Fig. 5):
**Exponential model** (Eq. 9)
with
$q_{\text{Plateau}}$ (real-time production (i.e., flux) of $CO_2$ in the probed chamber and connected rock space)
&
$CO_2{}_{\text{Excess}}$ ($CO_2$ stored initially in rock pores connected to the probed chamber; Eq. 12)

$pCO_2{}_{\text{Rock}}$

$q_{\text{Plateau}}$

$\frac{D(\text{CO2})}{\omega}$
(a diffusivity measure; Eq. 11)

$pCO_2{}_{\text{Rock}}$

$pO_2{}_{\text{Rock}}$

$CO_2{}_{\text{Excess}}$

$V_{\text{Rock pores}}$
(volume of the connected rock pore space; Eq. 13)

$\phi_{\text{Air-filled}}$ (porosity of the rock)

$V_{\text{Rock}}$
(volume of the connected rock; Eq. 14)

$z_{\text{Rock}}$ (weathering depth)

$\rho_{\text{Rock}}$ (rock density)

$S_{\text{Rock}}$
(surface of a theoretical outcrop with a comparable rock volume; Eq. 15)

$m_{\text{Rock}}$
(mass of the rock contributing $CO_2$ to the probed chamber; Eq. 16)

$\frac{D(\text{CO2})}{3}$

$j_{\text{O2}}$
($O_2$ flux into the rock space that emits the measured $CO_2$; Eq. 7,8)

Atm. $O_2$ measurement

*Legend*

Measurement

Output

Externally-derived input

[revised manuscript text omitted]